**mSystems**®

Ecological and Evolutionary Science

# Damage Repair versus Aging in an Individual-Based Model of Biofilms

Robyn J. Wright,[a,b,c,d,e] Robert J. Clegg,[b,c,d*] Timothy L. R. Coker,[b,c,d*] Jan-Ulrich Kreft[b,c,d]

aSchool of Life Sciences, University of Warwick, Coventry, United Kingdom

bSchool of Biosciences, University of Birmingham, Birmingham, United Kingdom

cInstitute of Microbiology and Infection (IMI), University of Birmingham, Birmingham, United Kingdom

dCentre for Computational Biology (CCB), University of Birmingham, Birmingham, United Kingdom

eDepartment of Pharmacology, Faculty of Medicine, Dalhousie University, Halifax, Nova Scotia, Canada

**ABSTRACT** The extent of senescence due to damage accumulation—or aging—is evidently evolvable as it differs hugely between species and is not universal, suggesting that its fitness advantages depend on life history and environment. In contrast, repair of damage is present in all organisms studied. Despite the fundamental trade-off between investing resources into repair or into growth, repair and segregation of damage have not always been considered alternatives. For unicellular organisms, unrepaired damage could be divided asymmetrically between daughter cells, leading to senescence of one and rejuvenation of the other. Repair of "unicells" has been predicted to be advantageous in well-mixed environments such as chemostats. Most microorganisms, however, live in spatially structured systems, such as biofilms, with gradients of environmental conditions and cellular physiology as well as a clonal population structure. To investigate whether this clonal structure might favor senescence by damage segregation (a division-of-labor strategy akin to the germline-soma division in multicellular organisms), we used an individual-based computational model and developed an adaptive repair strategy where cells respond to their current intracellular damage levels by investing into repair machinery accordingly. Our simulations showed that the new adaptive repair strategy was advantageous provided that growth was limited by substrate availability, which is typical for biofilms. Thus, biofilms do not favor a germline-soma-like division of labor between daughter cells in terms of damage segregation. We suggest that damage segregation is beneficial only when extrinsic mortality is high, a degree of multicellularity is present, and an active mechanism makes segregation effective.

**IMPORTANCE** Damage is an inevitable consequence of life. For unicellular organisms, this leads to a trade-off between allocating resources into damage repair or into growth coupled with segregation of damage upon cell division, i.e., aging and senescence. Few studies considered repair as an alternative to senescence. None considered biofilms, where the majority of unicellular organisms live, although fitness advantages in well-mixed systems often turn into disadvantages in spatially structured systems such as biofilms. We compared the fitness consequences of aging versus an adaptive repair mechanism based on sensing damage, using an individual-based model of a generic unicellular organism growing in biofilms. We found that senescence is not beneficial provided that growth is limited by substrate availability. Instead, it is useful as a stress response to deal with damage that failed to be repaired when (i) extrinsic mortality was high; (ii) a degree of multicellularity was present; and (iii) damage segregation was effective.

Address correspondence to Robyn J. Wright, robyn.wright@dal.ca, or Jan-Ulrich Kreft, J.Kreft@bham.ac.uk.

* Present address: Robert J. Clegg, Tessella, Abingdon, Oxfordshire, United Kingdom; Timothy L. R. Coker, HealthLumen, London, United Kingdom.

Development of a new individual-based model of aging biofilm cells with adaptive repair (cells sense and respond to their current intracellular damage levels) reveals the fundamental trade-off between investing resources into repair or growth.

KEYWORDS evolution, division of labor, mathematical modeling, aging, senescence, trade-offs

Senescence describes the deleterious effects of damage accumulation—commonly referred to as aging—that manifest in decreasing fecundity and/or increasing mortality with age. While senescence is all around us, it is not obvious why it has evolved in many taxa as it would appear to be detrimental to the fitness of individuals. Moreover, senescence is clearly evolvable as it varies hugely between species and is not universal (1). For example, several taxa of simple multicellular organisms can fully regenerate and for several taxa of complex multicellular organisms, fecundity does not simply decrease with age and/or mortality does not simply increase with age (2, 3). An evolutionary explanation for the various extents of senescence present in different organisms is challenging, particularly for unicellular organisms that divide apparently symmetrically such as most prokaryotes and some eukaryotes (here collectively referred to as "unicells"), in contrast to multicellular animals with a clear division of labor between germline and soma (4–6). However, the first single-cell study of division asymmetry in *Escherichia coli* highlighted that morphological symmetry does not exclude functional asymmetry; daughter cells inheriting the old cell pole were shown to grow a little slower than the mother cell, whereas the daughters with a new cell pole grew a little faster (7). Surprisingly, *Caulobacter crescentus*, the first bacterium studied in terms of senescence (8)—as it has substantial morphological and functional asymmetry in cell division—has been shown in more-recent high-throughput microfluidic studies to maintain a constant growth rate over cell divisions under benign conditions (9) and to divide protein aggregates symmetrically between mother and daughter cells (10).

Following the first single-cell studies that suggested the existence of senescence in unicellular prokaryotes (7, 8, 11) and unicellular eukaryotes (12), there has been a gold rush of studies eager to demonstrate senescence in further unicells, such as bacteria (13–15) and eukaryotic algae (16–19). The loss in fecundity (7) and increase in mortality (20) with age demonstrated in some of these unicells are, however, rather small effects compared with the resource limitations of growth and high external mortality in most environments. The effects are also much smaller than in the budding yeast, which has long been known to have a limited replicative life span (21), supported by several recent high-throughput single-cell studies (22–28). It may, however, be misleading to regard the budding yeast as a unicellular organism as wild relatives are capable of dimorphic growth (29) and domesticated strains rapidly evolve multicellularity (30, 31). Crucially, a number of recent experimental results have led to a reinterpretation of damage segregation as a stress response rather than as an evolved characteristic of growth under benign conditions (9, 10, 20, 32–37).

In concert with the gold rush for accumulation of experimental evidence of senescence in unicells, there has also been one of mathematical modeling studies eager to find evolutionary advantages of senescence. Some of these models did not include consideration of extrinsic mortality (38–40), although this favors rapid and early reproduction and thus tilts the evolutionary trade-off toward investment of resources into growth and reproduction, rather than maintenance and repair (2, 41). Some also did not consider repair as an alternative (39, 40, 42, 43) or did not consider the cost of repair (38).

Repair is present in all organisms studied, and evidence for the evolution of mechanisms that repair damage, such as misfolded and aggregated proteins (10), is beyond doubt. Of the few models that consider both extrinsic mortality and costly repair, the model reported by Ackermann et al. (2007) (44) was the first. It found that asymmetric damage segregation outperformed repair. In contrast, our previous study, Clegg et al. (2014) (45), found that repair was always beneficial whereas damage segregation was beneficial only in addition to repair and only if three conditions were fulfilled simultaneously: (i) the damage was toxic, (ii) damage accumulated at a high

mSystems®

rate, and (iii) repair was inefficient. The reason for this discrepancy in predictions could be pinned down as follows: in the study by Ackermann et al. (2007) (44), rather than growing, cells divided at fixed time intervals, while in that by Clegg et al. (2014) (45), cells grew by consuming substrate and divided when they reached a threshold size. This enabled both an immediate benefit of repair (increased growth rate of a less damaged cell leading to earlier division) and an immediate cost of repair (diverting resources away from growth to repair machinery). Overall, this made repair advantageous.

More-recent models that also considered repair produced similar conclusions (35, 46, 47). Vedel et al. (2016) (35), in particular, has advanced our understanding with an experimentally validated model that considers the fitness of the whole lineage. This helped to identify a positive-feedback loop that shifts growing lineages toward less-damaged cells, explaining how higher stress levels lead to higher levels of damage accumulation, which in turn leads to higher levels of damage segregation.

None of those studies considered the fitness effects of aging and repair in spatially structured environments such as biofilms, although biofilms are both prevalent in nature (with 40% to 80% of prokaryotic cells on Earth estimated to exist in biofilms [48]) and important for ecosystem function. For humans, biofilms have many advantages in biotechnology but also cause big problems in industry (49) and health (50). Biofilms are heterogeneous in both time and space (51), and cells growing within them are therefore exposed to varying, and often limiting, nutrient regimes (52). This leads to gradients in growth rate and the presence of an active layer in biofilms, as growth occurs only close to the nutrient-exposed surface of the biofilm due to slow nutrient diffusion (53). Slow growth within biofilms has also been shown to confer tolerance to damage-inducing agents, such as antibiotics (51, 54–56) and UV radiation (57, 58). It is therefore likely that these gradients of growth rate and stress could make the evolutionary benefits of aging and repair different from those seen in spatially uniform environments, such as chemostats. Moreover, biofilms have a clonal population structure unless the cells remain motile (59–61). This can have strong effects on the evolution of division of labor (62). Damage segregation can be seen as a division of labor akin to the germline-soma differentiation in multicellular animals (63). Thus, we hypothesized that biofilms might favor damage segregation over repair.

To test this hypothesis, we investigated the optimal strategy for dealing with nongenetic cellular damage in a generic unicell growing in a spatially structured, heterogeneous environment. Using the computational modeling platform iDynoMiCS (individual-based Dynamics of Microbial Communities Simulator) (64), we developed an agent-based model with adaptive repair (AR), whereby individual cells are able to sense and respond to their current intracellular damage levels, enabling an appropriate response to gradients of stress and damage in biofilms. Our simulations showed that adaptive repair—rather than damage segregation (DS) or a fixed rate of repair (fixed repair [FR])—was the optimal strategy for unicells growing in a biofilm, but only when the growth rate was limited by substrate supply and the rate of damage accumulation was proportional to the cells' specific growth rate.

## RESULTS

**Growth and repair of cells in iDynoMiCS.** Here, we investigated aging strategies of generic unicells, representing the relevant essence of all unicellular prokaryotes and eukaryotes. This means that we do not model a particular organism such as *E. coli* with specific, fixed characteristics and we do not model a particular repair mechanism for a particular type of damage as we are interested in the evolution of generic traits and strategies. We let cells grow in a biofilm environment simulated using the computational modeling platform iDynoMiCS (individual-based Dynamics of Microbial Communities Simulator) (64). Again, we are interested in a generic biofilm so we simulate cells growing into clusters on a flat, inert substratum with substrate diffusing into the biofilm from the surrounding liquid. In such a setup, a substrate concentration gradient forms, which leads to a gradient in growth rate and enables gradients of age, should they

occur, so this simple biofilm setup is sufficient for our current purpose. Aging is defined as accumulation of generic damage, rather than being chronological or based on the number of divisions (the budding yeast is the only known unicell with a limited replicative life span). Age is therefore a measure of the fraction of the biomass that is damaged. Detrimental effects of this age, or accumulated damage, are referred to as senescence. We refer to the entire biomass as "protein" as protein damage has been the focus of the literature, but the choice of word does not affect results. Note that we do not consider DNA damage and associated mutations, which are heritable and subject to natural selection, whereas natural selection does not directly act on the accumulation of misfolded proteins or the passive, inevitable damage segregation modeled here.

In iDynoMiCS, cells are modeled as individuals with embedded ordinary differential equations (ODEs) that are applied to each cell individually. Cells take in substrate and convert this to protein, which in turn leads to growth. This active, "autocatalytic" protein is damaged at a certain rate and becomes damaged, inactive protein. When cells reach a large enough size, they divide into two daughter cells with one of two division strategies. In symmetric division, each daughter cell inherits half of the damage. In asymmetric division, the old pole cell inherits all of the damage and the new pole cell inherits none and is therefore rejuvenated. In addition to these division strategies, cells have one of three repair strategies: no repair, fixed repair, or adaptive repair (Fig. 1). In our previous study (Clegg et al. [2014; 45]), simulations covered wide ranges of values of the aging relevant parameters (e.g., damage accumulation rate, investment into repair, and degree of asymmetry at division). We found that intermediate parameters always led to intermediate results, and we therefore did not test these again in all circumstances as we were interested only in qualitative results: the fitness ranks of the alternative strategies. A summary of all simulations performed can be found in Fig. S1 in the supplemental material.

**Characteristics of adaptive repair.** In fluctuating or spatially structured environments, such as biofilms, rates of growth and damage accumulation vary in time and space. An adaptive repair mechanism is therefore more appropriate than the fixed-repair strategy described in Clegg et al. (2014) (45) for constant and chemostat environments. For biofilms, we developed a new repair strategy where the fraction of newly synthesized protein, denoted by $\hat{\beta}$, allocated into repair machinery rather than into growth machinery depends on the current level of damage in the cell (Fig. 1). The idea was that a cell can sense and appropriately respond to its damage level. Adaptive repair is therefore meant to replace the previous strategy of a fixed allocation into repair machinery at an optimal level, $\beta$, which is appropriate only for constant or chemostat environments, where growth and damage accumulation rates are in a steady state, as reported previously by Clegg et al. (2014) (45). The purpose of this section is to examine the consequences of the new adaptive repair strategy and to test whether it is equivalent, in steady-state environments, to the previous optimal repair strategy.

**Consequences of adaptive repair for the fraction of repair protein in single cells.** The adaptive repair strategy leads to an allocation into repair that responds to current levels of damage and therefore lags the ideal level of repair machinery, unless damage levels reach a steady state due to symmetric division (Fig. S2). Since asymmetric division causes sudden changes in damage levels, the current investment into repair tracks the changing damage levels (Fig. 2A). Damage levels then change as a result of repair, which in turn changes allocation into repair. As a result, the repair machinery in asymmetrically dividing cells never reaches ideal levels, in contrast to symmetrically dividing cells (Fig. 2A; see also Fig. S2). To approach ideal levels of repair machinery more quickly, a high turnover of repair protein would be required. Since cellular proteins that are not involved in regulation have a long half-life of more than one generation (65), it seems more realistic to assume that repair protein does not turn over; turnover would also be costly and unnecessary. Investment into repair varies greatly over the cell cycle for cells with asymmetric segregation of damage but is

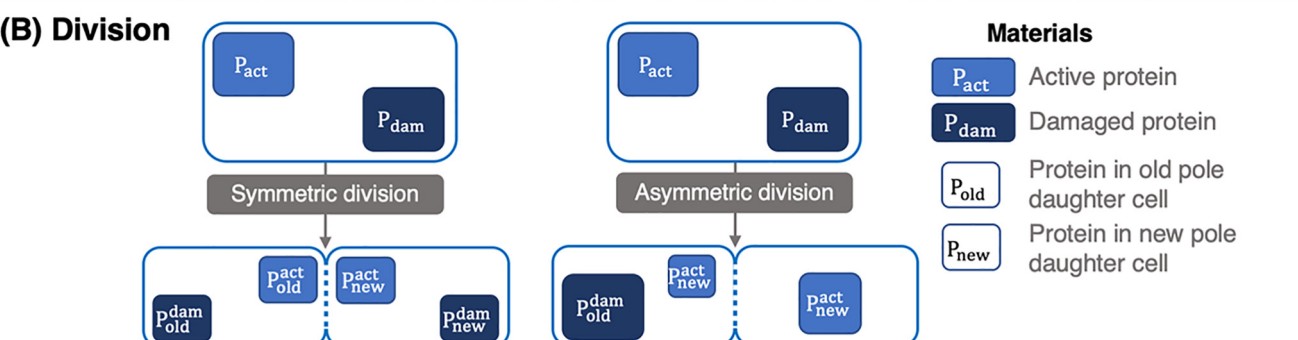

**FIG 1** Schematic overview of the individual-based model of cells growing by substrate consumption. Cells accumulate damage either at a fixed rate ($a = 0.1$ h$^{-1}$) or at a rate proportional to the gross specific growth rate ($a' = 0.22$ h$^{-1}/\mu_G$). (A) Damage either is not repaired ($\beta = 0$) or is repaired by dedicated repair machinery that does not contribute to growth, modeling a cost of repair. Investment into repair machinery takes place either at a fixed rate ($\beta = 0.07$) or in response to sensing the amount of damage in the cell (adaptive repair, $\hat{\beta}$). (B) Unrepaired damage is divided either symmetrically ($\alpha = 0$) into the daughter cells or asymmetrically ($\alpha = 1$) such that the old pole daughter cell receives all damage (damage segregation) and the new pole daughter cell is rejuvenated. The three repair strategies (A) and two division strategies (B) can be studied in any of six combinations: (i) asymmetric division (damage segregation) without repair (DS; $\alpha = 1$ and $\beta = 0$); (ii) asymmetric division with fixed repair (DSFR; $\alpha = 1$ and $\beta = 0.07$); (iii) asymmetric division with adaptive repair (DSAR; $\alpha = 1$ and $\hat{\beta}$); (iv) symmetric division with no repair (NR; $\alpha = 0$ and $\beta = 0$); (v) symmetric division with fixed repair (FR; $\alpha = 0$ and $\beta = 0.07$); and (vi) symmetric division with adaptive repair (AR; $\alpha = 0$ and $\hat{\beta}$). An overview of all simulations carried out, the key results obtained, and in which figures these results appear is shown in Fig. S1.

always at approximately the same level immediately before division. The investment into repair immediately before division is approximately the same as the fixed optimal investment into repair.

**Consequences of adaptive repair for growth rates of individual cells.** For asymmetric strategies, the cells with adaptive repair maintained the highest specific growth rate across consecutive divisions from approximately generation three onward (Fig. 2B). Without any repair, the specific growth rate declined rapidly toward zero. With fixed repair, specific growth rates likewise declined toward zero, albeit more slowly. For symmetric strategies, the specific growth rates were similar for all strategies while the

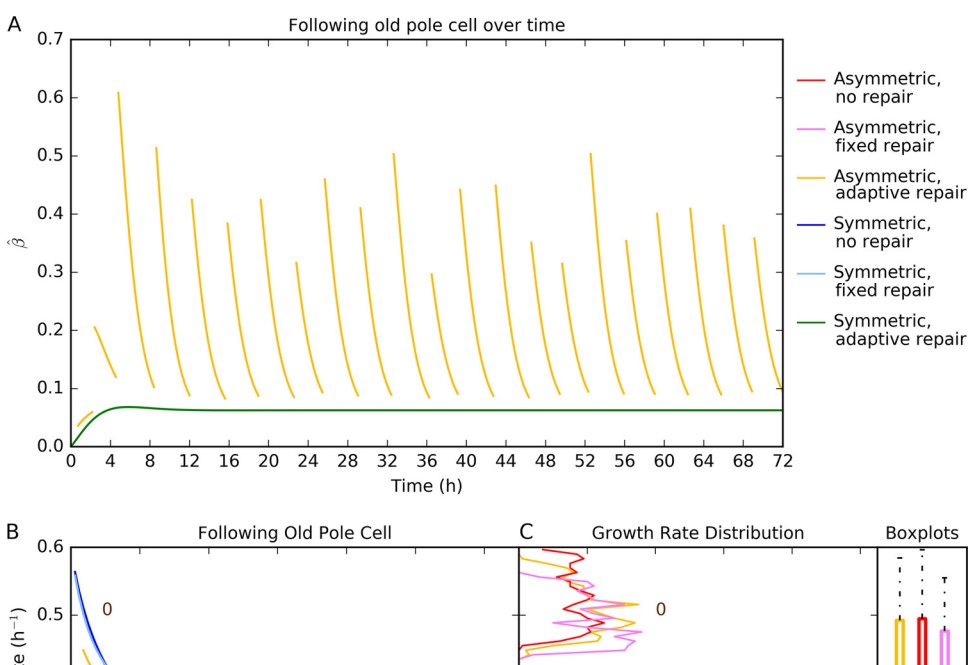

**FIG 2** Characteristics of strategies in a constant environment (no competition). (A) Investment into repair ($\hat{\beta}$ on the $y$ axis) for the new adaptive repair strategy following the old pole cell over many divisions. In asymmetric divisions, the old pole cell inherits all damage, leading to a jump in allocation into repair following division and then a steady decrease until the next division. In symmetric divisions, damage, and therefore investment into repair, reaches a steady state. (B) Specific growth rate of a single cell over consecutive cell divisions. Numbers in the panel label generations, and each generation is shown with a new line (for asymmetric strategies). The specific growth rate of an asymmetrically dividing cell with no repair (red, $\beta = 0$) drops quickly to zero. For a cell with fixed optimal repair (magenta, $\beta = 0.07$), the rate decreases more slowly over time but also reaches zero. For a cell with adaptive repair (yellow, $\hat{\beta}$ variable), the rate decreases only initially toward a seesaw pattern, as shown in panel A. Specific growth rates do not change at division for symmetric strategies (blue, cyan, and green), and there is no difference between daughter cells. Symmetric strategies show an initial decrease in specific growth rate before reaching a steady state, with similar values for fixed and adaptive repair and lower values without repair. (C) Distribution of specific growth rates in populations at steady state (snapshot taken at 100 days) for asymmetrically dividing cells. Specific growth rates of cells with adaptive repair are between those with fixed repair and those without repair. The medians and interquartile ranges for adaptive repair and fixed repair are close and higher than for symmetrically dividing cells. Data are reproduced from Fig. 4A and B in reference 45 with permission, with the addition of the new adaptive repair strategy. The specific growth rate was 0.6 h$^{-1}$, and the damage accumulation rate was 0.22 h$^{-1}/\mu_G$ and was dependent upon the specific growth rate for all strategies. Replicate simulations are similar; see the figures in the file at https://figshare.com/articles/figure/Damage_repair_versus_aging_in_an_individual-based_model_of_biofilms_supplementary_file_2/12883832.

damage levels were low. Later, cells with repair maintained substantially higher specific growth rates than cells without repair (Fig. 2B).

**Consequences of adaptive repair at the population level.** A comparison of age and total biomass data for each cell in a population shows the distribution of the levels of damage and how close the cells are to division (division is triggered when cells reach a total biomass threshold; Fig. 3). For asymmetric strategies, adaptive repairers did not reach the high damage levels (old ages) seen with other strategies. For symmetric

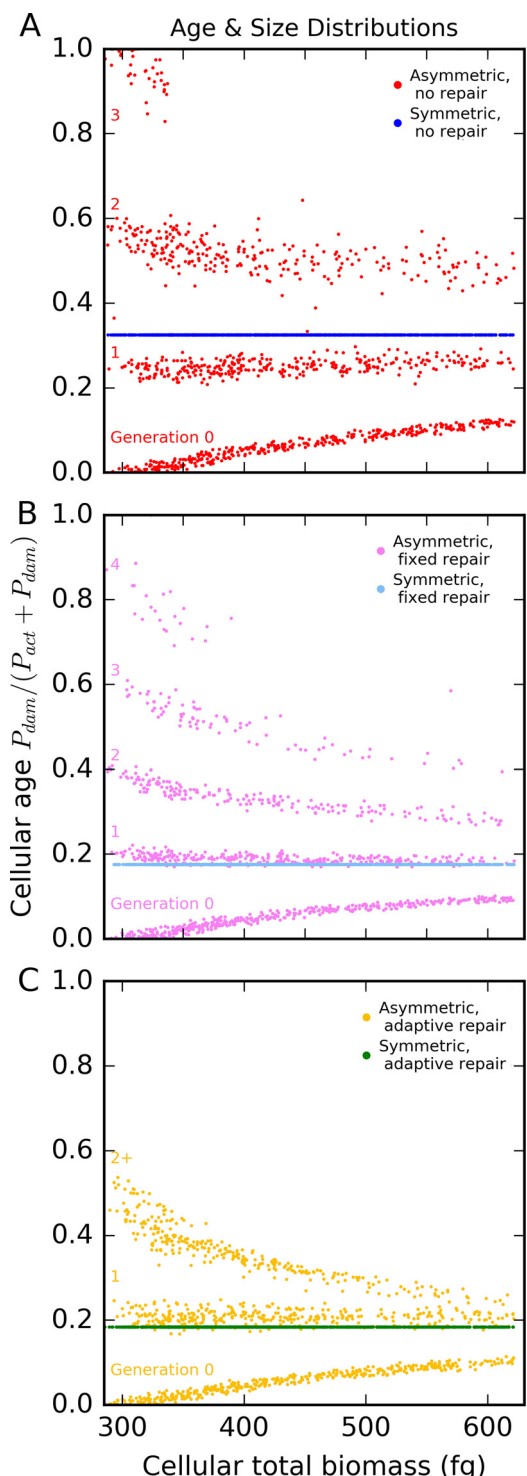

**FIG 3** Age and size distributions of populations in a constant environment (snapshot at 100 days, after reaching steady state). (A) Strategies without repair. (B) Strategies with fixed repair. (C) Strategies with adaptive repair. For symmetric division, cells without repair were substantially older than cells with fixed or adaptive repair, which were of similar ages. For asymmetric division, the age of cells with adaptive repair stays well below the maximum age of one and reaches this level from generation three onward so there are fewer very old cells than are seen with fixed repair or without repair. Repair reduces the age of older cells over the cell cycle. The largest cells are approaching the division threshold, $P_{div}$, of 620 fg. Data are reproduced from Fig. 4C and D in reference 45 with permission, with the addition of the new adaptive repair strategy. The specific growth rate was 0.6 h$^{-1}$, and the aging rate was 0.22 h$^{-1}/\mu_G$ and was dependent upon the specific growth rate for all strategies. Replicate simulations are shown in the file at https://figshare.com/articles/figure/Damage_repair_versus_aging_in_an_individual-based_model_of_biofilms_supplementary_file_2/12883832.

strategies, adaptive repairers were marginally older than fixed repairers but much younger than cells without repair. Growth rates in populations of asymmetrically dividing cells had a multimodal distribution with marked differences between generations (Fig. 2C). There were fewer cells with very high and very low specific growth rates in populations using either fixed or adaptive repair. The growth rate distribution of the new adaptive repair strategy was between the distributions seen with the fixed and no-repair strategies. The medians for each population confirm that the population-specific growth rates were highest for cells using the fixed-repair strategy, though those seen with the adaptive repair strategy were only slightly lower and showed less variation between cells (Fig. 2C). The symmetrically dividing cells all had the same specific growth rate as the single cells represented in Fig. 2B, with the fixed repairers again having a slight specific growth rate advantage. In summary, the new adaptive repair strategy led to growth rates that were very similar to those seen with fixed repair at the population level, but at the individual level, there were fewer old cells.

**Competitions of aging strategies in constant and chemostat environments.** Competitions are unambiguous and unbiased ways to measure fitness holistically rather than using arbitrary fitness functions or definitions that ignore that fitness emerges from interactions between competing strategies (45). In the constant environment, cells were removed at random, which modeled extrinsic mortality, and the strategy that was left at the end was assessed as having won the competition. In the chemostat environment, the cells were likewise removed randomly, but they also competed for the substrate that entered the environment at a given rate. This means that the lineages that produced fewer offspring per elapsed time at the current concentration of substrate were washed out. In other words, the cells with the highest specific growth rate (depending on the substrate concentration) would emerge as the winners (on average, as removal is stochastic). The damage segregation without repair (DS) strategy quickly lost against each of the repair strategies (the fixed-repair [FR] and adaptive-repair [AR] strategies) in both environments (Fig. 4). The winner took much longer to emerge in the competitions between the two repair strategies, and there were large fluctuations in the biomass ratios over the course of the simulations. Fixed repairers (FR) had an advantage in the constant environment, whereas in the chemostat, the adaptive repairers (AR) won in the end. Hence, the new adaptive repair strategy was slightly fitter than the fixed-repair strategy in those natural environments that are better approximated by chemostats than by constant environments, such as systems that are mixed on a reasonably short time scale and that receive resource inputs and experience removal of biomass by various means. However, more environments are spatially structured and are therefore better modeled by biofilms, so we turn to those in the next section.

**Generating realistic biofilm structures in the absence of damage accumulation.** We first identified which parameter set would give rise to typical, rough biofilm structures with "finger" formation (66–68), rather than flat biofilms, so that we could then study aging in biofilms using realistic biofilm structures (Fig. S3; see also the figures in the file at https://figshare.com/articles/Damage_repair_versus_aging_in _biofilms-File_S1_pdf/11520534). These simulations were without damage accumulation or repair. The substrate concentration in the bulk liquid was varied in order to change the dimensionless group $\delta^2$ that quantifies the extent to which biofilm growth is limited intrinsically (growth-limited regime, high $\delta^2$) or limited extrinsically by diffusional mass transport into the biofilm (transport-limited regime, low $\delta^2$; see Text S1 in the supplemental material for more information on $\delta^2$). Since the levels of biofilm roughness at the end of the simulations were not significantly different among the three $\delta^2$ regimes tested, we decided to continue our simulations with only one value for $\delta^2$, i.e., the intermediate value where $\delta^2 = 0.0069$.

**Biofilm simulations with constant damage accumulation rate.** For the biofilm simulations with a constant damage accumulation rate, we compared asymmetric damage segregation without repair (DS) and symmetric damage segregation with adaptive (AR) or fixed (FR) repair.

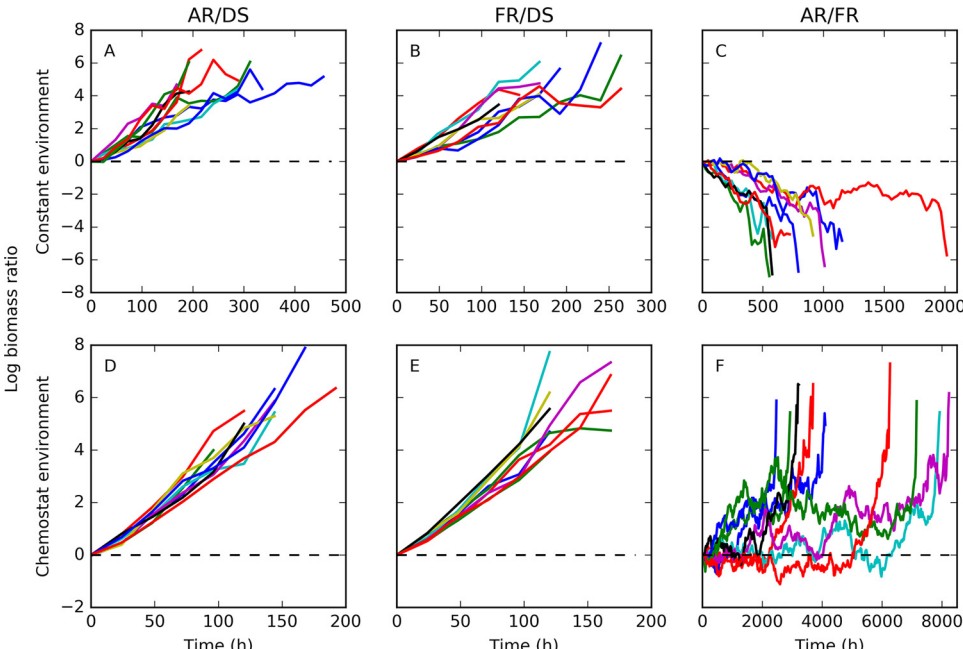

**FIG 4** Pairwise competitions of aging strategies in constant and chemostat environments. AR, adaptive repair without damage segregation; FR, fixed optimal repair without damage segregation; DS, damage segregation without repair. Different colors represent replicate simulations ($n = 10$ for each competition), and the log biomass ratio shown in each panel is indicated in the heading (i.e., AR/DS, FR/DS, and AR/FR for panels A and D, panels B and E, and panels C and F, respectively). Both repair strategies (AR and FR) were substantially fitter than DS in either environment (A, B, D, and E; proportion test, $P = 0.00195$ for all). The two repair strategies were closer in fitness, as seen in Fig. 2B, and it took more than an order of magnitude longer before the final outcome was clear. FR was slightly fitter than AR in the constant environment (C; proportion test, $P = 0.00195$), as expected from the results shown in Fig. 2B, but AR was slightly fitter than FR in the chemostat environment (F; proportion test, $P = 0.00195$). The maximum specific growth rate was 0.6 $h^{-1}$, and the damage accumulation rate was 0.22 $h^{-1}/\mu_G$ and was dependent upon the specific growth rate for all strategies.

Repair of damaged material is assumed to require resources, e.g., energy and some new material to replace the damaged parts of the old material. These resources are assumed to be supplied by endogenous metabolism of cellular material rather than by the substrate, as the latter is not always available. As a result, converting damaged material into undamaged, active material comes at a loss of biomass (we assume a loss of 20% for reasons given in the previous study by Clegg et al. [2014; 45]). This loss leads to shrinking of cells (since the density of cells is assumed to be constant), unless cells grow sufficiently fast to compensate, which is not the case in the lower layers of a biofilm. Shrinking does not affect fitness in constant and chemostat environments as only the number of organisms matters, but the shrinking of cells under adaptive repair strategy conditions had profound effects on biofilm structure and reduced the fitness of this strategy in biofilms (Fig. 5). In these competitions, the winning strategy depended upon the initial cell density. When the initial cell density was low, it appears that the winning strategy depended more upon the initial, random placement; cells of any strategy that happened to be placed furthest away from the cells of the other strategies tended to win. When the initial cell density was higher, the winning strategy was more dependent upon the fitness of that strategy. Because the cells of the adaptive repair strategy shrank so much more than the cells of the other strategies, this effect was much greater for them.

How much of the disadvantage of adaptive repairers was due to shrinking can be seen by comparing simulations performed with shrinking cells to identical simulations where, for the sake of comparison, the lost material is assumed to continue to take up volume (i.e., the lost material has no mass but keeps its original volume, dubbed "styrofoam"). In the simulations without styrofoam, AR was fitter than DS only at initial densities of above 8 cells, FR was fitter than DS at initial densities above 4 cells, and FR

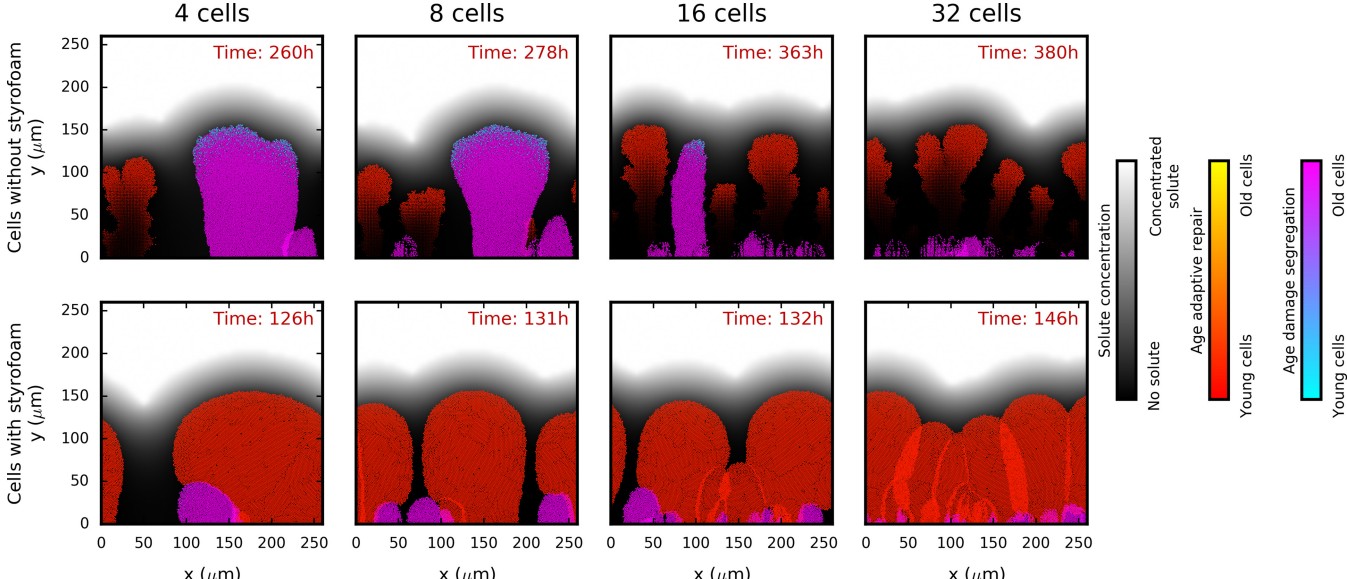

**FIG 5** Damage segregation strategies (DS, cool colors) versus adaptive repair strategies (AR, warm colors) in biofilms. Adaptive repairers are strongly affected by shrinking (top row) as they are much fitter under conditions in which they are assumed not to shrink, i.e., when the material lost due to repair is replaced with inert and massless material of the same volume ("styrofoam," bottom row). Initial cell density increases from left to right (4, 8, 16, or 32 total cells). The biofilm structures shown have all reached a height of 154 $\mu$m. Cells are colored by age, with different color gradients for each species. The maximum specific growth rate was 1.2 h$^{-1}$, and the damage accumulation rate was 0.1 h$^{-1}$ (not proportional to the specific growth rate) for all. Results of replicate simulations and of simulations which competed DS and AR against the fixed optimal repair strategy (FR) and controls are shown in figures in the file at https://figshare .com/articles/Damage_repair_versus_aging_in_biofilms-File_S1_pdf/11520534. See Fig. S4 for time courses of log biomass ratios. For simulations without styrofoam, DS tended to be fitter than AR at initial densities of 4 and 8 cells, but AR was fitter at the higher initial densities of 16 and 32 cells (top row). For competitions between DS and FR, FR was fitter at a density of 8, 16, or 32 cells (see figures in the file at https://figshare.com/articles/Damage_repair_versus _aging_in_biofilms-File_S1_pdf/11520534; see also Fig. S4). In the simulations with styrofoam, AR and FR were always fitter than DS (bottom row; see figures in the file at https://figshare.com/articles/Damage_repair_versus_aging_in_biofilms-File_S1_pdf/11520534; see also Fig. S4). For competitions between AR and FR, FR was fitter in simulations without styrofoam, while AR was fitter in simulations with styrofoam. Control simulations that competed two cells with the same strategy always led to no clear winner.

was fitter than AR in all simulations (cell numbers always refer to total number of cells). In the simulations with styrofoam, the results were much clearer; the cells subjected to AR and FR are fitter than the damage-segregating cells and AR was fitter than FR, regardless of the cell density at the beginning of the simulation (Fig. 5; see also Fig. S4).

The results caused by shrinking under our study conditions were thought to be unrealistic. Cells have not been observed to shrink considerably, unless they have been starved for long periods of time, and biofilm structures such as those represented in Fig. 5, with a vanishing base due to endogenous metabolism, have not, to our knowledge, been observed and would be mechanically unstable in the presence of shear (69). Either shrinking must therefore be very limited in real cells or the assumption that nongrowing cells accumulate damage at the same rate as rapidly growing cells (which would cause only the nongrowing cells to shrink, as the growing cells can make up the lost volume) must be wrong. We decided to avoid this unrealistic shrinking result by assuming that cells that do not grow also do not accumulate damage, i.e., that damage is a by-product of metabolism. The damage accumulation rate was therefore assumed to be proportional to the specific growth rate for individual cells (and was matched to the previous damage accumulation rate; Fig. S5), to allow for the very low rates of growth of cells below the active layer of the biofilm (the active layer is shown in Fig. 6). This means that shrinking is not abolished but that slowly growing cells accumulate damage and shrink at a lower rate because repair, leading to shrinking, is less necessary. See Materials and Methods for further explanation of this proportional damage accumulation rate. We therefore continued with a damage accumulation rate that was proportional to the gross specific growth rate (0.22 h$^{-1}$/$\mu_G$) and also applied this to the earlier constant and chemostat environment simulations.

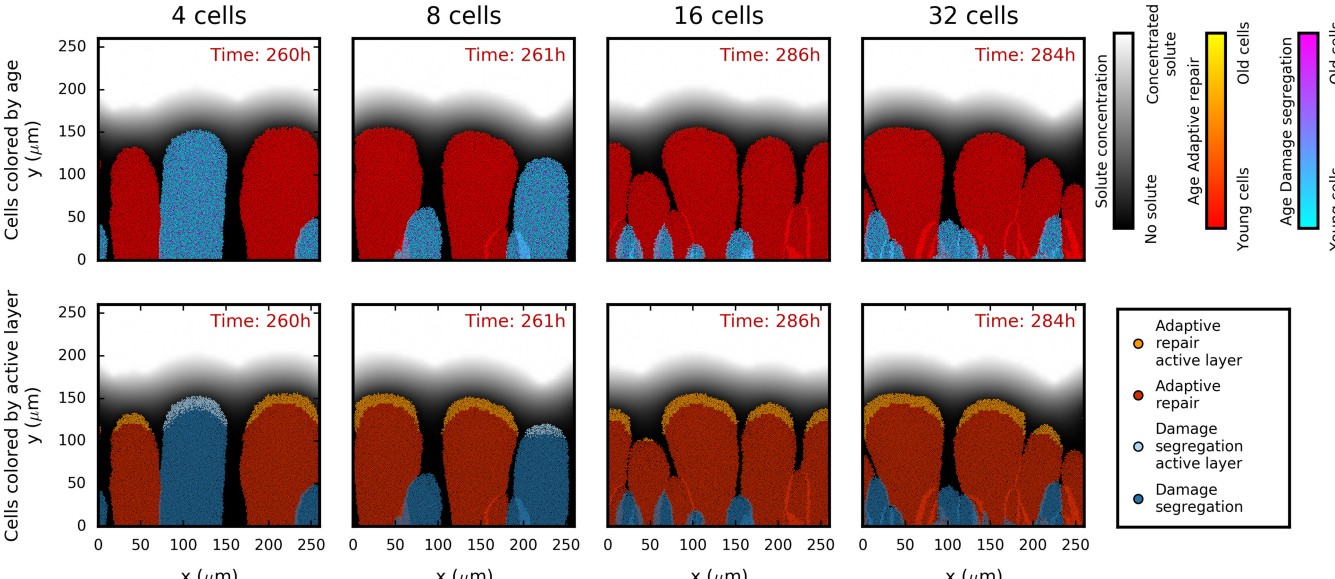

**FIG 6** Damage segregation versus adaptive repair competitions in biofilms where the damage accumulation rate is proportional to the specific growth rate. Initial cell density increases are shown from left to right (4, 8, 16, and 32 cells). Adaptive repair becomes more advantageous with greater initial cell density. In the top row, cells are colored by age, while in the bottom row, cells are colored more brightly if they are in the active layer of rapidly growing cells, defined as cells with a specific growth rate within 5% of the highest at the indicated time point. The biofilm structures shown have all reached a height of 154 $\mu$m. The maximum specific growth rate was 1.2 h$^{-1}$, and the damage accumulation rate was set at 0.22 h$^{-1}$/$\mu_G$. Repeat simulations ($n = 50$) are shown in figures in the file at https://figshare.com/articles/Damage_repair_versus_aging_in_biofilms-File_S1_pdf/11520534, and control competitions are shown alongside this figure in the file at https://figshare.com/articles/figure/Damage_repair_versus_aging_in_an_individual-based_model_of_biofilms_supplementary_file_2/12883832.

AR performed better than FR in the biofilms with styrofoam (Fig. 5), and we therefore focus on the two main alternative strategies, AR and DS, in the following section.

**Biofilm simulations where the damage accumulation rate is proportional to the specific growth rate.** When the damage accumulation rate was proportional to the specific growth rate, AR was more competitive than DS (Fig. 6; see also Fig. 7). The higher the initial cell density, the stronger the advantage of AR and the earlier that strategy won. At the highest initial cell density (32 cells), AR won in all 50 replicate simulations ($P = 0.00$). At the lowest cell density (4 cells), AR won in the majority of the simulations (31 versus 19) but this difference was not statistically significant ($P = 0.119$). The advantage of adaptive repair became statistically significant at initial cell densities of eight or higher (see Table S1A in the supplemental material).

We decided to use the log biomass ratios as the best measure for statistical comparison of fitness after comparing the performances of a range of different metrics when the two strategies analyzed were identical (controls in Table S1A and in the file at https://figshare.com/articles/figure/Damage_repair_versus_aging_in_an_individual-based_model_of_biofilms_supplementary_file_2/12883832). The ideal fitness measure should not be time dependent, such that running simulations longer would not change outcomes. As the trends of the log biomass ratios in Fig. 7 show, using biomass at the end of the simulations (approximately 250 h) was appropriate since the qualitative outcomes would not have changed had the simulations continued. Moreover, the fitness metric combined with a particular method of significance testing should never result in a statistically significant result when the two competitors are identical, as was the case with the final growth rate (Table S1A). The final growth rate was therefore not a suitable measure. Both biomass and population size could be used; they are strongly coupled, but the biomass changes are smoother than the cell numbers; hence, biomass was chosen as the fitness measure.

The biofilms formed from these equal mixtures of DS and AR, initially placed randomly on the substratum surface, showed the spatial distribution of the strategies

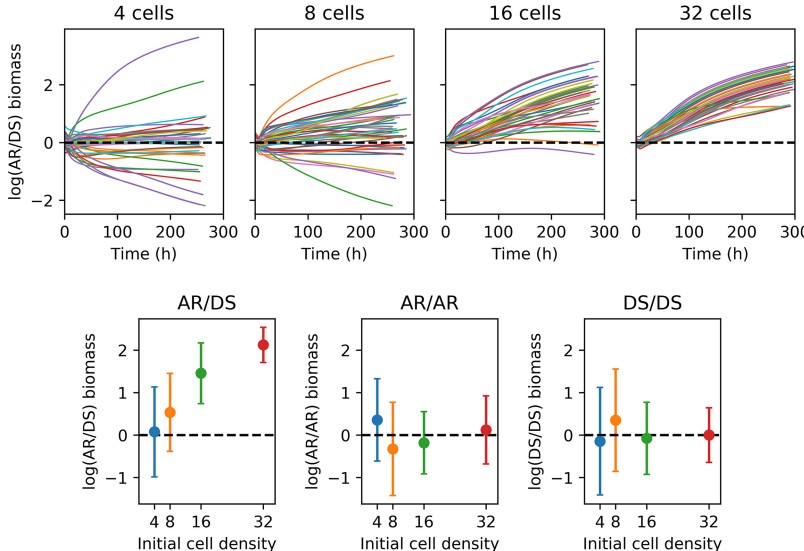

**FIG 7** Time courses of log biomass ratios for 50 replicate biofilm competitions between adaptive repair (AR) and damage segregation (DS) strategies and their controls. The higher the initial cell density, the more advantageous adaptive repair (AR) became compared with damage segregation (DS). At the start, 4, 8, 16, or 32 cells were randomly placed on the surface. Results are shown using log biomass ratios to make the horizontal line at log(ratio) = 0 (i.e., ratio = 1) a symmetry axis. The top row shows time courses for AR versus DS competitions, while the bottom row shows mean log(ratio) values with standard deviations for all simulations, including controls, taken when biofilms had reached 154 $\mu$m (approximately 250 to 300 h). Different colors denote replicate simulations (top row) or different initial cell densities (bottom row). Statistics for these competitions are shown in Table S1, and biofilm structures are plotted in Fig. 6 (see also figures in the file at https://figshare.com/articles/Damage_repair_versus_aging_in_biofilms-File_S1_pdf/11520534). Control time courses are shown in the file at https://figshare.com/articles/figure/Damage_repair_versus_aging_in_an_individual-based_model_of_biofilms_supplementary_file_2/12883832. The maximum specific growth rate was 1.2 h$^{-1}$, and the damage accumulation rate was set at 0.22 h$^{-1}$/$\mu_G$ and was dependent upon the specific growth rate for all strategies.

and the age and activity of the cells (Fig. 6). First, it can be seen that there was limited mixing of cells where neighboring fingers touched (lines of different color standing out), consistent with our previous work (70, 71). This is due to the assumption that cells embedded in the biofilm matrix are not motile. Second, when finger-like cell clusters were similarly separated from other clusters, the two strategies resulted in similar heights, suggesting that shrinking was negligible (Fig. 6; 4-cell initial density). This was despite repair leading to loss of material and cellular volume as growth could compensate for any losses when the rate of damage accumulation was proportional to the growth rate. Third, the effect of increasing the initial cell density can also be seen. When there were only a few cells randomly placed on the substratum surface, it was likely that one cell would be further away from competing cells than any of the other cells. This lineage could therefore grow without much competition and become the highest finger, producing the highest biomass regardless of which strategy it followed. Hence, results at low cell density were more dependent on the random initial cell positions determining the distance to competitors than on competitiveness. The higher the initial density, the less the results were influenced by the stochastic initial attachment of cells on the surface. In these cases, AR had the advantage. Finally, a comparison of the distribution of young versus old cells with the distribution of active versus inactive cells (Fig. 6) shows that the cells subjected to AR were all young throughout the biofilm whereas the cells subjected to DS included a mixture of young (rejuvenated) and old (damage-keeping) cells throughout the cell clusters, regardless of height. Note that cells were active only when they were at the top of their biofilm finger and when this finger was at least (about) as high as the neighboring fingers because only then did they receive sufficient substrate diffusing in from the bulk liquid, which was separated from the biofilm by a concentration boundary layer (66–68). Hence, whether a strategy

mSystems®

still had cells in the active layer was determined by whether it still had cells growing close to the maximum biofilm height and not by how many cells it had that were very young.

**The impact of altering key growth and aging parameters on competition outcomes.** We carried out a sensitivity analysis by varying maximum specific growth rate ($\mu_{max}$), biomass growth yield (efficiency of growth; $Y_\mu$), yield of repair (efficiency or repair; $Y_r$), division threshold (cell size at division; $P_{div}$), substrate concentration in the bulk liquid supplying the biofilm ($S_{bulk}$), substrate affinity ($K_s$), and growth rate proportional aging rate ($a'$). These simulations competed AR against DS, and the winner was determined by the long-term trend of the log biomass ratio, as described above, using one higher and one lower value for each parameter (a total of 14 additional parameter values, $n = 45$ simulations). The higher and lower values were based on the highest and lowest experimentally validated values in the literature (mostly for *E. coli*) or were half/double or 10 times higher/lower than the original values, depending on which was appropriate for each parameter (Table S1B). AR remained a fitter strategy than DS with three exceptions: yield of repair, substrate concentration, and substrate affinity (Fig. 8; see also Fig. S6). As expected, reducing the efficiency of repair reduced the advantage of repair. (DS won 2 of 3 simulations when repair yield was reduced from 0.8 to 0.444 g active biomass g$^{-1}$ damaged biomass such that repair of biomass was no longer more efficient than *de novo* biomass synthesis; in other words, repair yield became equal to growth yield). This was in line with previous results from constant-environment simulations (45). More revealing were the effects of increasing the bulk substrate concentration from 0.003556 to 0.03556 g liter$^{-1}$, where DS won 3 of 3 simulations, and of increasing the substrate affinity (lowering $K_s$ values from 0.00234 to 0.0000234 g liter$^{-1}$), where DS won 3 of 3 simulations (Fig. 8; see also Fig. S6 and the figures at https://figshare .com/articles/Damage_repair_versus_aging_in_an_individual-based_model_of_biofilms _supplementary_file_2/12515526). We identified a combination of mechanisms (explained in detail below) that led to DS becoming the fitter strategy under these environmental conditions where growth rate was less dependent on substrate concentration.

First, we need to explain how positive feedback arises in biofilm growth generally (regardless of aging) and then how this can be enhanced by a growth rate split due to damage segregation. Generally, positive feedback arose since cells grew fastest at the highest location in the biofilm where the substrate concentration was highest (Fig. 9) as this was where the substrate diffusing into the biofilm from the liquid above first arrived. As the highest cells grew fastest, they were producing more biomass and therefore more offspring than other cells in the biofilm, adding more height to the highest location, which amplified the height difference (Fig. S7; see also the movies of biofilm development at https://figshare.com/articles/Damage_repair_versus_aging_in _an_individual-based_model_of_biofilms_supplementary_file_2/12515526) and in turn increased the growth advantage of the cells at the top—representing positive feedback. For a more rigorous treatment of this effect, known as "fingering instability," see Dockery and Klapper (2002) (67).

Now consider that the damage segregation strategy split the population into rejuvenated, fast-growing cells and consecutively older, slower-growing cells (Fig. 9C). This amplified the positive feedback described above when the young cells at the top grew under near-substrate-saturating conditions where the absolute difference in growth rate from the repairers—which, while young, grew slower as they shunted resources into repair machinery—was highest. We refer to this as positive-feedback enhancement. At near-substrate-saturating concentrations, the absolute growth rate difference was primarily age dependent, whereas at lower substrate concentrations, the growth rate was proportional to the substrate concentration, reducing the absolute difference in growth rate due to age. Thus, at near-substrate-saturating concentrations (substrate concentration above $K_s$), changes in substrate concentration had little effect on growth rate (Fig. 9A and B) and DS won because the absolute difference between the growth rates of the recently rejuvenated cells (arising from damage segregation)

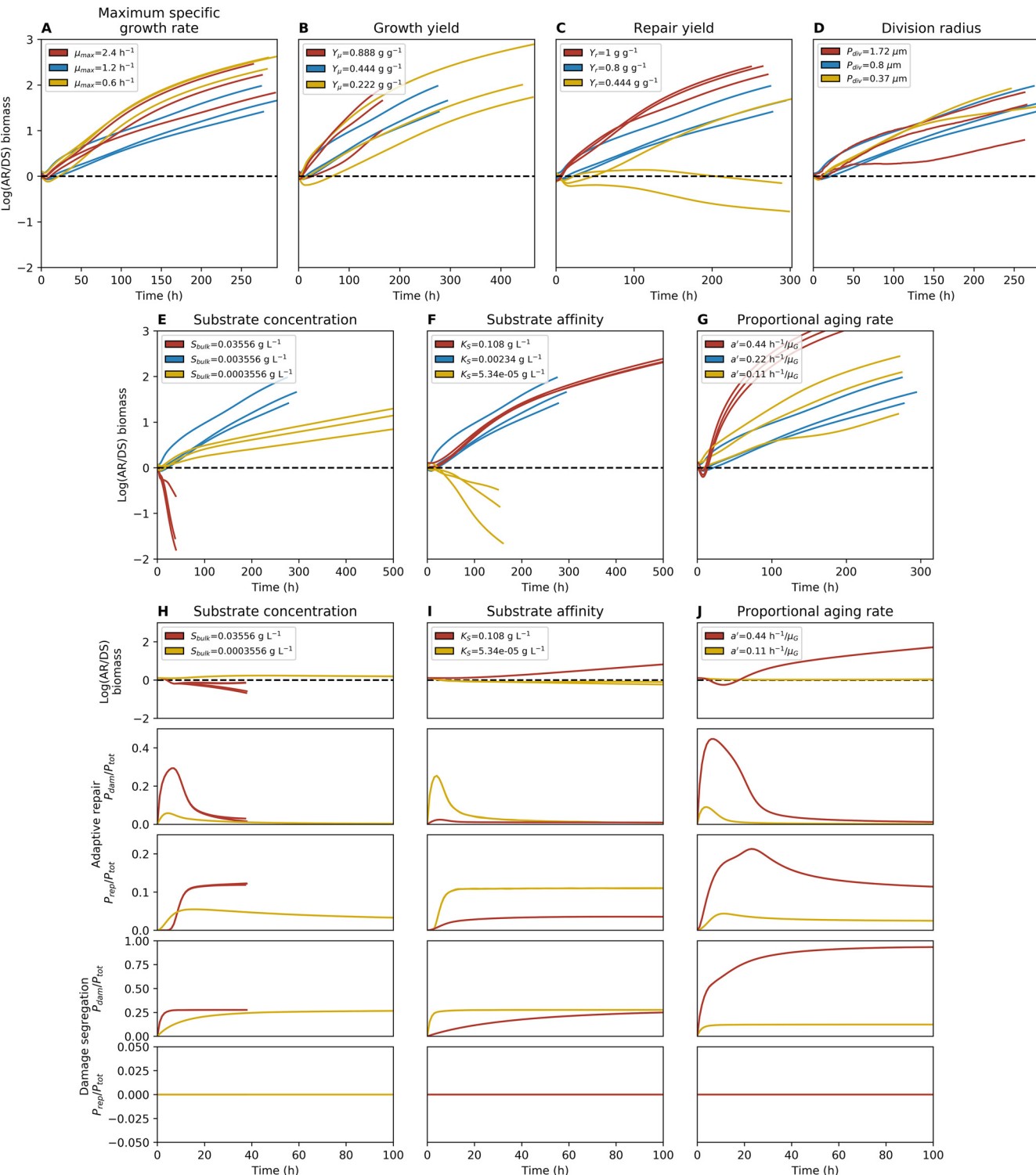

**FIG 8** Competitions between adaptive repair and damage segregation strategies in biofilms with varied key growth parameters. (A to G) Time courses of log biomass ratios are shown for three replicates for the original values used (blue lines; the same simulations are used for each varied parameter), higher values (red lines), and lower values (yellow lines). Cells were initially placed at random ($n = 16$ per strategy), and simulations were stopped when a maximum biofilm height of 154 $\mu$m or a time of 500 h was reached. (H to J) Time courses of log biomass ratios as well as the total population fraction of protein that is damaged or devoted to repair are shown for two replicates of the higher values (red lines) and lower values (yellow lines) for each of substrate concentration, substrate affinity, and proportional aging rate. Cells were initially placed in two side-by-side blocks ($n = 16$ per strategy), and simulations were stopped when a maximum biofilm height of 154 $\mu$m or a time of 100 h was reached.

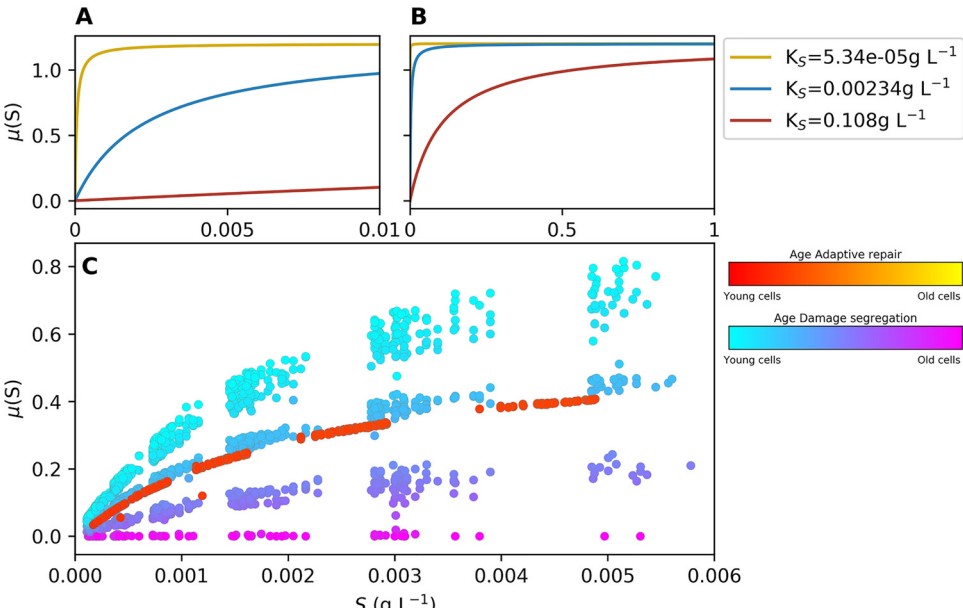

**FIG 9** (A and B) Maximum specific growth rate, $\mu(S)$, as a function of substrate concentration (S, g liter⁻¹) between 0 and 0.01 (A) or between 0 and 1 (B) for the three values used for substrate affinity, $K_S$. (C) Specific growth rate versus substrate concentration for individual cells in a simulated competition between DS and AR after 10 h of biofilm growth at intermediate $K_S$ and high bulk substrate concentration. Each point corresponds to one cell and is colored according to the age of the cells. Cells were initially placed in two side-by-side blocks ($n = 16$ per strategy; this biofilm is shown in Fig. S7). Note that local substrate concentrations reported in panel C are approximate since the substrate concentration is resolved in a grid with resolution of $(4 \times 4)$ μm² whereas cell locations are precise.

and the also young but repair-invested AR cells was highest (Fig. 9C). This is why this positive-feedback enhancement was able to become strong enough under substrate saturation conditions to compensate for the disadvantage of producing older cells by damage segregation, which grew more slowly than the AR cells (Fig. 9).

Thus, we suggest it was a combination of two mechanisms (positive feedback and split in growth rates leading to positive-feedback enhancement) that gave DS an advantage under the environmental condition of nearly saturating levels of substrate at the top of the biofilm. The substrate concentration being at close to saturating level in the active layer at the top of the biofilm could be explained in either of two ways: (i) the substrate concentration was very high in the bulk liquid supplying the biofilm or (ii) the substrate affinity was so high that substrate concentration was close to saturation even at low concentrations (Fig. 9A and B). We propose that this explains why DS won under the two conditions of high substrate concentration in the bulk liquid and very high substrate affinity (low $K_S$) of the cells (Fig. 8).

We also identified two other effects that give DS an advantage, and yet these do not explain the benefit observed under substrate saturation conditions. The first is the effect of volume loss due to repair; as repair is not 100% efficient, it leads to loss of some of the mass of the damaged material upon repair and to a reduced volume of the cells. This effect can be artificially disabled by assuming that the loss of mass does not lead to a loss of volume (converting the lost mass into the massless volume that we dubbed "styrofoam"). When repairers produced styrofoam to avoid volume loss, the simulations still showed an advantage, albeit reduced, of DS at substrate-saturating conditions (Fig. S7). The second is an initial advantage for the DS strategy, present in all simulations—with the effects being more substantial in some—as can be seen in the log biomass ratios and repair investment over time shown in Fig. 8. There are actually two separate reasons for this. The first is that AR cells were initialized without any repair machinery, so they started life unprotected and accumulated damage, which then triggered investment into repair and, consequently, reduction of damage. This initial

disadvantage was an artifact of the simulation setup. Its effect was short-lived and had reduced impact when strategies were spatially more separate, growing side by side and competing only where they met. Since these showed the same trends as the random initial placements (Fig. 8), this effect was not important. The second is that when few cells were initially placed on the surface, there was little substrate consumption and high substrate concentrations were therefore available to all cells—the same situation as described above for the combination of positive-feedback enhancement and substrate saturation. However, this effect did not last as the substrate became more limiting when the number of cells consuming substrate increased (see movies of biofilm growth at https://figshare.com/articles/Damage_repair_versus_aging_in_an _individual-based_model_of_biofilms_supplementary_file_2/12515526). Also, this effect had little effect on the outcome of competitions when the competing strategies were operating side by side (Fig. 8), in contrast to the positive-feedback enhancement.

**The prevalence and abundance of repair genes in prokaryotes.** Since our simulations suggested that repair was advantageous in well-mixed environments and also under almost all biofilm growth conditions (when growth was limited by substrate concentration), we asked whether repair genes were present in all unicells as expected. We identified the genes for the repair machineries that are used for disaggregating and refolding denatured proteins in well-characterized microorganisms (Table S1C). We searched for these genes in the annotated KEGG ortholog lists from 10 "model" prokaryotic genomes with the largest number of assemblies in the NCBI database (Table S1D) as well as in 20,000 microbial genomes from the Integrated Microbial Genome (IMG) database (72). The searches revealed that all 10 of the prokaryotic model genomes and 19,983 of the 20,000 IMG genomes included at least one repair gene (Fig. S8). Further investigation and reannotation of those 17 genomes that were apparently without repair genes found repair genes in 15 of them (Table S1E). The two genomes that remained without repair genes were assigned taxonomically to *Staphylococcus* and *Proteobacteria* (lower taxonomic classification was not possible) and were found to be only 42% and 47% complete, respectively. We expect that complete genomes for these two organisms would contain repair genes, as they are present in the other 19,998 genomes.

## DISCUSSION

Here, we investigated two alternative strategies for use by unicells to deal with intracellular damage: segregation and repair. Due to a trade-off between investment of cellular resources into repair machinery versus the growth machinery that is fundamental to all living organisms, repair is costly and therefore not obviously beneficial. This study was the first to our knowledge to compete these strategies in biofilms, representing spatially structured environments. We began by developing a new model for adaptive repair of cellular damage, whereby cells are able to sense and respond to current damage levels by investing into repair machinery accordingly, enabling cells to deal with spatial and temporal changes of conditions in biofilms. We found that under almost all conditions tested, adaptive repair was fitter than asymmetric segregation of damage at division (Fig. 4, 7, and 8). We also compared adaptive repair with our previous fixed optimal repair strategy (45)—in well-mixed environments where both strategies are suitable. Surprisingly, although cells with the adaptive repair mechanism had slightly lower growth rates than those with fixed optimal repair at the level of the individual (Fig. 2B), they outperformed them in the chemostat environment where there was competition for resources (Fig. 4). The likely reason for this result is that with adaptive repair, there were fewer cells growing at the lowest rate (Fig. 2C); as cells grew exponentially, any slight growth rate advantage or disadvantage would increase over time, before stochastic removal from the chemostat.

When we initially applied the adaptive repair model to growth in biofilms, we were surprised that the bases of biofilm fingers of adaptive repairers were disappearing (Fig. 5). Since we assume that converting damaged material into new material incurs a loss of 20% to account for the energy requirement of repair, the slowly growing cells

at the base of the biofilm are not able to compensate for this 20% loss of biomass due to repair with new growth. Therefore, repairers shrink. This could be considered a biofilm-specific disadvantage of repair. However, we think that the idea of such extensive shrinking is unrealistic. First, to our knowledge, extensive shrinking of the volume of starving or dormant cells other than that occurring by cell division has not been observed, potentially due to the murein sacculus maintaining cell shape. Second, the idea of biofilm structures with completely disappearing bases is not realistic as the slightest shear stress would detach these structures (69, 73, 74). Moreover, one would expect that the higher the rate of metabolism, such as that represented by protein folding or respiratory electron transport, the higher the chance of damage arising, such as by protein misfolding or generation of reactive oxygen species (19, 75, 76). Indeed, organisms that grow more rapidly have been shown to also accumulate damage more rapidly (46, 77) and can have a higher rate of mortality (37). Therefore, we decided to choose the simplest assumption, i.e., that the rate of damage accumulation should be proportional to the specific growth rate of individual cells. In this case, biofilm fingers of adaptive repairers no longer had shrinking bases (Fig. 6) and thus performed better than the damage segregators (Fig. 7; see also Table S1A in the supplemental material), as may be expected when cell death is predominantly intrinsic, rather than extrinsic as in the constant and chemostat environments. Unfortunately, there are no empirical studies comparing the benefits of damage segregation versus repair in biofilms, highlighting the dearth of studies of aging in the most common habitat of microorganisms.

The evolved extent to which unicells segregate damage asymmetrically varies substantially between species. This poses the question of whether this is due to differences in the mechanism of cell growth and division, differences in the "life span" of their habitats, or differences in the degree to which these unicells are actually multicellular with a clearer division of labor between germline and soma. The budding yeast *Saccharomyces cerevisiae* was the first unicell shown to age (21) and, in fact, is the only unicell for which the evidence of senescence—in the form of active segregation of damage upon cell division and a limited replicative life span—under benign conditions, rather than as a stress response, remains strong (9, 10, 20, 32–37, 78–81). Its habitat is very rich in sugars but is short-lived (82), and we argued previously (45) that investing resources into repairing the cell rather than reproduction is less advantageous when the habitat is (reliably) transient. However, the fission yeast *Schizosaccharomyces pombe* lives in the same kind of habitat (32) but does not age under benign conditions (see below). Also, the bacterium *Caulobacter crescentus* lives attached to surfaces that are decaying or consumed by zooplankton and are therefore similarly transient, and yet the evidence for senescence in *C. crescentus* has dwindled since our 2014 publication (9, 10, 83). This suggests that our previously proposed explanation—that morphologically asymmetric cell division and high external mortality due to short-lived habitats are necessary and sufficient conditions to evolve senescence in unicells—needs to be revised as these conditions appear necessary but not sufficient. We propose that a third condition, nascent multicellularity, must also be met (see below).

Recent studies of the fission yeast, following single cells for many generations, provide clear evidence that asymmetric damage segregation does not occur under benign conditions. Instead, it appears to be a stress response to deal with misfolded proteins, which aggregate and then fuse into fewer and larger aggregates, facilitating the segregation of the damage into one aged daughter cell with a reduced growth rate and higher mortality (32, 33, 84). Spivey et al. (2017) (36) found that cell death in the fission yeast was actually not preceded by the characteristics of senescence. Nakaoka and Wakamoto (2017) (37) also found no increase of mortality with age but instead found mortality to increase with growth rate. Moreover, they found that aggregates can be lost from old pole cells during division so that they can rejuvenate. Remarkably, oxidative stress reduced growth rate only transiently and protein aggregates present in the cell after stress did not affect growth. Apart from fission yeast, a recent study of *C. crescentus* by Iyer-Biswas et al. (2014) (9) found a trend of decreasing fecundity over

generations only at 37°C, the highest temperature used. This could be considered a heat shock given the typical lake temperatures where *C. crescentus* lives. Moreover, Schramm et al. (2019) (10) found no evidence for asymmetric segregation of protein aggregates, despite the morphologically asymmetric cell division in *C. crescentus*. In *E. coli*, the extent to which a cell segregates damage asymmetrically at division tends to increase with the severity of the environmental stress that cells are exposed to (35). In *Mycobacterium tuberculosis*, segregation is critical for recovery from stress that results in damaged protein that cannot be repaired (34). An entirely different issue is survival during starvation, where senescence has recently been proposed as an adaptive strategy based on the finding that starving *E. coli* cells, like nonstarving humans, follow the Gompertz law of mortality (85). However, the Gompertz law has been shown to arise from a variety of processes, including tumor growth, growth of batch cultures, and genetic or acquired susceptibilities to death (86–91), so no mechanistic conclusions can be drawn from finding that mortality follows the phenomenological Gompertz law.

Considering all evidence, the budding yeast appears to be the only studied unicell where senescence occurs in the absence of stress. However, neither its asymmetric cell division mechanism of budding nor its living in transient habitats is sufficient to explain this difference. It seems to us that the common view of the budding yeast as a unicell may be mistaken and that the missing ultimate reason why budding yeast senesces is that it is, to some extent, a multicellular organism. First, the monophyletic yeast lineage (*Saccharomycotina*) is a branch of the *Ascomycota*, a division of fungi whose members have many mycelial forms with multicellular hyphae, so yeasts have a multicellular heritage (92). Second, yeast can easily evolve toward multicellularity when cluster formation is selected (30, 31). Third, while many lab strains are mutants with mutations in a gene required for filamentous growth to facilitate genetic analysis, wild strains of yeast are dimorphic, with a unicellular yeast and a pseudohyphal multicellular form under starvation conditions (29). Thus, the "unicellular" yeast is at the cusp of multicellularity. We propose that the combination of the budding mechanism of asymmetric growth and division, dispersal between transient habitats, and nascent multicellularity represents the ultimate reason that the budding yeast is the exception that evolved senescence as part of the normal life cycle rather than as a stress response.

Our study had several limitations. First, we have focused on the effect of damage on growth rate rather than mortality. This is partly for simplicity and partly because some studies have shown an increase of mortality with age (93) whereas others suggested that mortality is random rather than an event that occurs more frequently with increasing age (32, 36, 37). Second, we neglect damage that had not been repaired before segregation or that would be prohibitively expensive to repair, since the work of Lin Chao's group has covered this well. They have shown that damage segregation under constant environmental conditions leads to separate steady-state levels of damage in cells of both old and young lineages, meaning that growth rate and mortality of cells do not change over divisions (42, 43, 94–97). (Since there is no trend of deterioration and the replication of young and old lineages does not fit a germline-soma distinction, it is probably better to refer to this as damage homeostasis rather than senescence.) Third, we avoided specific assumptions of mechanisms of damage repair or segregation that are organism specific as these are subject to evolution and our interest is the evolution of general strategies. Fourth, we have simplified the biofilm system to consist of growth on flat, inert surfaces without detachment. While our rough, finger-like biofilms capture typical aspects of biofilm structure, many processes and potential structures could not be covered in this study as the number of possible combinations is huge. Nonetheless, our study was the first to cover the extremes, ranging from a perfectly mixed chemostat to a simple biofilm without any mixing (no motility of cells and only diffusive transport of substrate through boundary layer and biofilm).

Since the results were, essentially, the same for both extremes, we had expected that our findings showing repair being more beneficial than damage segregation would hold for all other environments, particularly given that we found genes for repair

in 99.9% of prokaryotic genomes (see Fig. S8 in the supplemental material). However, our sensitivity analysis showed that under some biofilm conditions, for example, when the substrate affinity or substrate concentration is high enough, damage segregation is the fitter strategy (Fig. 8; see also Fig. S7). It is likely that most biofilms in the environment grow under conditions of strong nutrient limitation, growing just enough to compensate for losses and therefore remaining approximately in a steady state. However, near hot spots of substrate supply, there could be regions of high substrate concentrations in the liquid flowing along the biofilm boundary layer before the substrate becomes gradually depleted by consumption in the biofilm, leading to oligotrophic conditions further away from the hot spot or where the hot spot turned cold. Near these hot spots, damage segregation is predicted to be advantageous as it allows for the highest possible growth rates prior to substrate limitation setting in, representing a case similar to the previously discussed transient habitats.

In conclusion, our models' predictions that repair of damage is more advantageous than segregation of damage under a wide range of conditions are confirmed by the experimental literature showing that repair mechanisms are present in all model unicells and that senescence due to damage segregation does not occur under benign conditions when repair can handle any damage effectively. However, senescence provides fitness advantages (in the absence of stress) if the following necessary and sufficient conditions are met in combination: (i) the presence of an active, robust, and efficient mechanism for damage segregation, such as budding; (ii) living in a habitat strongly favoring early reproduction (such as a "short-lived" habitat, high extrinsic mortality, or growth racing to the top of a biofilm when lack of substrate limitation facilitates the positive-feedback enhancement); and (iii) the presence of a degree of multicellularity that enables a germline-soma division of labor that is not just about damage segregation. Otherwise, senescence is advantageous only as a stress response to deal with damage that failed to be repaired, which explains why repair genes are present in virtually all unicells. Here, we have expanded the scope of this prediction substantially from previous work performed in constant and dynamic but spatially uniform environments by exploring biofilms as exemplars of spatially structured systems, thought to harbor the majority of microbes in the environment. We found a number of additional mechanisms that operate in biofilms due to the gradients of substrate concentration, growth, and aging rates present as well as the clonal population structure. In contrast to our original hypothesis that the clonal population structure in biofilms would favor senescence, we found that repair is also better than damage segregation in biofilms, provided growth is limited by substrate availability. This is likely to hold for environments that are between the extremes of well-mixed environments (chemostats) and unmixed environments (biofilms of immotile cells).

## MATERIALS AND METHODS

We followed the standard ODD (overview, design concepts, and details) protocol for describing individual-based models to facilitate comparison and review (Grimm et al. [98, 99]). (See Fig. S1 in the supplemental material for an overview of the aims of and simulations performed for each section of this study.)

**Purpose.** The purpose of this study was to determine whether segregation or repair of damage is the optimal strategy in response to cellular damage in spatially structured systems such as biofilms and for generic unicellular organisms (unicells). In order to do this, we expand upon our previous work in spatially uniform systems (constant and chemostat environments; Clegg et al. [2014; 45]), by introducing a strategy for adaptive damage repair and further subdividing biomass into four rather than two components. All simulations described here were performed using the free and open-source modeling platform iDynoMiCS (individual-based Dynamics of Microbial Communities Simulator; version 1.5) (64). Within iDynoMiCS, sets of ordinary differential equations (ODEs) are used to describe metabolic reactions within each cell, allowing cells within the simulations to respond appropriately and individually to gradients within their environment, for example, in substrate concentrations. Here, this allowed for the heterogeneity in the conditions experienced by organisms growing at the bottom of a biofilm compared with those at the top. The heterogeneity comes as a consequence of substrate consumption by cells in the biofilm and of substrate diffusing into this sink from the surrounding bulk liquid as a source (a diffusion reaction system).

mSystems®

**TABLE 1** All symbols, variables, and parameters used in this study

| Symbol | Name | Unit |
|---|---|---|
| $a$ | Damage accumulation (aging) rate (constant) | $0.1\ h^{-1}$ |
| $a'$ | Damage accumulation (aging) rate proportional to gross specific growth rate | $0.22\ h^{-1}/\mu_G$ |
| $\beta$ | Investment in repair as a proportion of the total protein (fixed) | Dimensionless, 0.07 |
| $\hat{\beta}$ | Adaptive investment in repair as a proportion of the total protein | Dimensionless, between 0 and 1 |
| $\mu_{max}$ | Maximum specific growth rate (Koch & Wang, 1982 [105]) | $1.2\ h^{-1}$ |
| $\mu(S)$ | Specific growth rate as a function of substrate concentration | $h^{-1}$ |
| $K_S$ | Monod (half-saturation) constant or substrate affinity (Koch & Wang, 1982 [105]) | $0.00234\ g\ liter^{-1}$ |
| $P_{g,a}$ | Protein invested into growth, active | fg |
| $P_{r,a}$ | Protein invested into repair, active | fg |
| $P_{g,d}$ | Protein invested into growth, damaged | fg |
| $P_{r,d}$ | Protein invested into repair, damaged | fg |
| $P_{act}$ | Active protein ($P_{g,a} + P_{r,a}$) | fg |
| $P_{dam}$ | Damaged protein ($P_{g,d} + P_{r,d}$) | fg |
| $P_{gro}$ | Protein invested into growth ($P_{g,a} + P_{g,d}$) | fg |
| $P_{rep}$ | Protein invested into repair ($P_{r,a} + P_{r,d}$) | fg |
| $P_{tot}$ | Total protein ($P_{g,a} + P_{r,a} + P_{g,d} + P_{r,d} = P_{act} + P_{dam}$) | fg |
| $P_{div}$ | Threshold cell radius triggering division, equivalent to a volume of 2 $\mu m^3$ (45) | $0.8\ \mu m$ |
| $Z$ | Cellular age, or damaged protein as a proportion of total protein $\left(\dfrac{P_{dam}}{P_{tot}}\right)$ | Dimensionless, between 0 and 1 |
| $\alpha$ | Degree of asymmetry, or damage segregation | Dimensionless, between 0 (fully symmetric) and 1 (fully asymmetric) |
| $\theta$ | Baby mass fraction; proportion of protein inherited by the new pole cell (normally distributed with mean 0.5 and SD 0.025) | Dimensionless, between 0 and 1 |
| $Y_\mu$ | Growth yield, the efficiency of converting glucose to active protein (Neijssel et al., 1996 [106]) | $0.444\ g\ g^{-1}$ |
| $Y_r$ | Repair yield, the efficiency of converting damaged protein to active protein (45) | $0.8\ g\ g^{-1}$ (assumed) |
| $r(P_{act},P_{dam})$ | Rate of repair | $h^{-1}$ |
| $S$ | Substrate concentration | $g\ liter^{-1}$ |
| $S_{in}$ | Substrate concentration in the inflow (45) (used in the constant and chemostat environments) | 0.00234 (constant) or 0.00324 (chemostat) $g\ liter^{-1}$ |
| $S_{bulk}$ | Substrate concentration in the bulk liquid (used in the biofilm environment) | 0.014222, 0.000889, or 0.003556 $g\ liter^{-1}$ |
| $D$ | Dilution rate in the chemostat (45) | $0.3\ h^{-1}$ |
| $\rho$ | Biomass density (dry mass); assumed to be lower in biofilms due to the presence of extracellular polymeric substances (EPS) | 290 (constant and chemostat [107]) or 201 (biofilm) $g\ liter^{-1}$ (assumed) |
| $b_L$ | Boundary layer thickness (unless otherwise stated) | $48\ \mu m$ (assumed) |
| $\delta^2$ | Ratio of maximum substrate transport to maximum substrate consumption rate | Dimensionless, 0.0017, 0.0069, or 0.028 (assumed) |
| $\sigma_f$ | Absolute deviation of biofilm front points from the mean front position; a measure of biofilm roughness | $\mu m$ |
| $h_{max}$ | Maximum biofilm thickness (unless otherwise stated) above which cells would be removed (here, the simulation was stopped when this height was reached) | $154\ \mu m$ (assumed) |
| $D_G$ | Diffusivity of growth substrate (glucose) | $6.7 \times 10^{-10}\ m^2\ s^{-1}$ (108) |
| $S_f$ | Shove factor (109) | 1.10 (assumed) |

**State variables and scales. (i) Growth parameters.** The growth parameters (Table 1) used are those for *Escherichia coli*, where available, and are the same as those used by Clegg et al. (2014) (45). Note that "protein" is representative of the whole biomass.

**(ii) Environments.** Three environments were used for this study: constant, chemostat, and biofilm.

*(a) Constant environment.* The substrate concentration is kept constant, and the population size is kept to 1,000 as new cells randomly replace existing cells.

*(b) Chemostat environment.* The simulation domain behaves like a chemostat of size 1 mm³. A chemostat is a well-mixed system where fresh resources constantly flow in and cells and leftover resources constantly flow out, at the same dilution rate $D$ (0.3 $h^{-1}$; Table 1).

*(c) Biofilm environment.* Only two-dimensional simulations were used, to simplify analysis; however, the addition of a third dimension would not be expected to change results because the two horizontal dimensions are equivalent (71). The domain size is (256 by 256) $\mu m^2$, and the spatial grid for solving the diffusion reaction equation has a resolution of (4 by 4) $\mu m^2$. The glucose concentration in the bulk liquid connected to the biofilm domain is kept constant throughout.

**Length of time simulated.** Constant and chemostat environment simulations were run for a maximum length of 500 days. Single-species simulations were run for 500 days, and competition simulations were stopped earlier if only one species remained in the simulation domain. Some single-species, single-cell simulations were run for only 3 days when the purpose was to follow one old pole cell over sufficient generations. Biofilm simulations were run until a maximum biofilm height of 154 $\mu m$ was reached.

**Process overview and scheduling. (i) Growth, damage accumulation, and repair.** Cell growth is exponential, as growth rate is proportional to the current mass of the cell (but see the details of age

dependence described below). Cells divide once their total protein threshold, $P_{div}$ (Table 1), is reached (randomized by drawing from a normal distribution with given mean $\pm$ standard deviation of $0.5 \pm 0.025$). Cells are made of two types of biomass, referred to as protein: protein invested into growth machinery and protein invested into repair machinery. Such protein may be either active or damaged (Table 1). As cells grow, they make active protein that is invested into growth. Active protein is damaged in one of two ways: at a constant rate ($a$), as in our previous work (45), or at a rate that is proportional to the cellular specific growth rate ($a'$). In cells that possess the ability to carry out repair, damaged protein is converted back to active protein, at a rate proportional to both the concentration of damaged protein and the concentration of active repair protein with rate constant $\beta$, but with an efficiency, or repair yield ($Y_r$), of 80% (these processes are summarized in Fig. 1).

Individuals can differ in their strategies for dealing with cellular damage, but strategies are inherited and do not evolve. There are two cell division strategies: cells divide either symmetrically or asymmetrically. Here, we looked only at complete symmetry or asymmetry as partial asymmetry always gave intermediate results in our previous study (45). In an asymmetrically dividing cell, the "old pole" daughter cell inherits all damage (up to capacity), while the "new pole" daughter cell inherits none and is therefore rejuvenated. There are three repair strategies: (i) no repair; (ii) investing into repair machinery at a fixed fraction ($\beta$) of newly formed biomass (the optimal fixed fraction of investment as a function of damage accumulation rate was previously determined for chemostats; 45); and (iii) adaptive investment into repair machinery ($\dot{\beta}$) depending upon the current levels of damage within the cell. This hypothesis of adaptive repair strategy is new to studies that model damage accumulation and repair in unicells and was developed to account for the different specific growth rates of cells in a biofilm, which change in time and space. Altogether, there were six combinations of division and repair strategies used in this study (Fig. 1). All six combinations of division and repair strategies were used for initial, single-strategy investigations (i.e., Fig. 2), but only FR, AR, and DS were used for competitions.

**(ii) Mortality (intrinsic and extrinsic).** In all environments, cells may be considered dead when their age reaches 1, signifying that there is no longer any active protein within the cell (intrinsic mortality). Such "dead" cells are assumed to remain physically intact and to continue to occupy space (only relevant for biofilms) since cell wall degradation is presumably a slow process (taking many residence times in the chemostat and longer than the times we simulate in biofilms).

*(a) Constant environment.* A cell is removed at random each time a division occurs (extrinsic mortality).

*(b) Chemostat environment.* Cells are removed randomly with the outflow from the chemostat at the dilution rate (extrinsic mortality).

*(c) Biofilm environment.* Cells are not removed from the simulation for the reasons given in the fitness section (no extrinsic mortality). Rather than removing the cells when the maximum biofilm height is reached (simulating detachment), we stop the simulation.

**(iii) Biofilm structure.** The formation of biofilm structure was investigated in the absence of damage accumulation and repair in order to find conditions that would give typical biofilm structures, e.g., a smooth biofilm, an intermediate biofilm, and a rough biofilm. Earlier models have found that rougher and more-finger-like biofilms tend to be produced when nutrient availability is limiting growth (66, 67, 100). This can be achieved, e.g., by reducing the substrate concentration in the bulk liquid $S_{bulk}$ or increasing the thickness of the boundary layer $b_L$, which both have a direct effect on the flux with which substrate diffuses into the biofilm and therefore on the thickness of the actively growing layer (68). Dimensionless groups have previously been introduced to explain the combined effects of $S_{bulk}$ and $b_L$ and other parameters on biofilm structure (see Text S1 in the supplemental material for further explanation of these). Here, we used the following equation:

$$\delta^2 = \frac{S_{bulk} D_G Y_\mu}{\mu_{max} \rho b_L^2}$$

where $D_G$ is the diffusion coefficient of the growth substrate, $Y_\mu$ is the growth yield, $\mu_{max}$ is the maximal specific growth rate, and $\rho$ is the biomass density. In order to obtain biofilm structures that were smooth, intermediate or rough, $b_L$ was kept constant at 48 $\mu$m and $S_{bulk}$ took three different values (in g liter$^{-1}$): 0.014222 (smooth, $\delta^2 = 0.028$), 0.003556 (intermediate, $\delta^2 = 0.0069$), and 0.000889 (rough, $\delta^2 = 0.0017$).

**(iv) Time.** Time is discrete. Since diffusion and biochemical reactions are on a faster timescale than growth and cell division, the diffusion-reaction equation is solved while the biomass distribution is fixed. Once the substrate concentration field is updated, it is kept fixed while the agents are stepped so they can grow and divide. Both the diffusion-reaction and agent time steps were set to be the same at 0.01 h$^{-1}$ in constant and chemostat environments and 0.05 h$^{-1}$ in biofilm environments, where rates are lower.

**(v) Event scheduling.** The simulation is entirely time stepped rather than event driven. The order in which agents are called in each time step is randomized.

**Design concepts. (i) Adaptation.** Within this model, only those cells with the adaptive repair mechanism are able to adapt to their environment. These cells are able to sense their current cellular damage levels and invest into producing repair protein appropriately. Repair protein is assumed to be stable rather than turned over. In other words, repair protein is not converted back into growth protein if it is no longer needed.

**(ii) Fitness.** Fitness is an emergent property and not defined by a fitness function. In the constant and chemostat environments, the fittest strategy is determined by competition as the only strategy remaining in the simulation domain. In biofilm simulations, the fittest strategy was determined by comparing population sizes, growth rates, and the log biomass ratios between the strategies. We have avoided

detachment in these biofilm simulations as a metacommunity model would be required to simulate fitness as an emergent property when cells detach from a biofilm patch as other patches would need to be present for colonization by cells detached from the focal patch.

**(iii) Observation.** Data are written to xml files at user-defined intervals. These data include the state of the environment, such as solute concentration fields and summary data on the population (e.g., number of cells born or removed at each time step), as well as lists of individual cells with all of their properties, such as position, size, growth rate, and investment into each protein type. Since the entire state of all agents and the environment is saved, simulations can be restarted with this output.

**Initialization. (i) Environment.** For the constant and chemostat environments, simulations are initiated with 1,000 cells (500 per strategy). Biofilm simulations are initiated with 4, 8, 16, or 32 cells (2, 4, 8, or 16, respectively, of each strategy), which are placed randomly on the substratum surface.

**(ii) Input.** Almost all system and agent parameters are specified in an xml input file called the 'protocol' file. Example input files can be found at https://github.com/R-Wright-1/iDynoMiCS_1.5.

**Details (submodels). (i) Mathematical skeleton.** The following equations are for modeling growth, damage accumulation, and repair of individual cells. They are ordinary differential equations (ODEs). Their solution depends on conditions prescribed at one end of the interval of interest, usually the initial conditions (Lick, 1989 [101]).

**(ii) Individual model equations.** The population is not modeled directly, but summary statistics are gathered and rates summed over all individuals. The substrate consumption rates of all individuals are gathered and summed, and this total rate of substrate consumption enters the standard equation for chemostat substrate dynamics (+inflow − outflow − consumption). Note that the net specific growth rate of an individual is also the sum of the rates of change for all four components of the cell. We give the differential equations for the change of the cell's components below. Individuals do not have access to population level information, and their behavior depends only on local conditions.

The biofilm environment consists of substrate concentration fields and a representation of the current biofilm structure (substratum surface, biofilm, biofilm boundary-layer interface, and boundary-layer bulk-liquid interface). The environment is modeled as a continuum using partial differential equations (PDEs) to describe the diffusion of substrate and rates of substrate uptake (or product secretion) by the cells. For this purpose, the distributions of cellular masses and substrate consumption (reaction) rates are mapped to the grid used for solving the PDEs. The reaction diffusion PDEs are solved with a multigrid algorithm (see Lardon et al. [2011; 64] for more details).

Since adaptive repair has a variable fraction of repair machinery, each of the previously used $P_{act}$ and $P_{dam}$ factors had to be split into two fractions as follows:

$P_{g,a}$, $P_{r,a}$, $P_{g,d}$, and $P_{r,d}$, referring to growth machinery, active; repair machinery, active; growth machinery, damaged; and repair machinery, damaged, respectively (as in Table 1).

Thus, the total "protein" of the cell (representing all biomass) is $P_{tot} = P_{g,d} + P_{r,d} + P_{g,a} + P_{r,a}$.

We assume that damage is always toxic, i.e., that the specific growth rate, due to some inhibitory effect of damaged material, decreases with the fraction of damaged protein, $Z$, equivalent to the age of the cell:

$$Z = \frac{P_{g,d} + P_{r,d}}{P_{tot}}$$

Toxic damage led to more-pronounced differences between the strategies in our previous study but did not change the fitness ranking of strategies, apart from one case, at the lowest damage accumulation rate and only in the constant environment, where the differences between strategies were minute (45).

In cells of all strategies, growth of active "protein" depends on substrate concentration $S$ following Monod kinetics:

$$\mu(S) = \frac{\mu_{max} S}{K_S + S} \tag{1}$$

Where repair of damage takes place, the rate of repair is Michaelis-Menten-like and proportional to damaged protein and active repair protein (see Clegg et al. [2014; 45] for further explanations):

$$r(P_{act}, P_{dam}) = \frac{\tilde{\beta} P_{act} P_{dam}}{\tilde{\beta} P_{act} + P_{dam}} \tag{2}$$

where $\tilde{\beta} P_{act}$ represents the proportion of active biomass that is dedicated to repairing damaged biomass and a placeholder for the value actually used depending on the repair strategy. For fixed repair, it becomes the fixed fraction $\beta$ of active protein $P_{act}$ that is repair machinery $\tilde{\beta} P_{act} = \beta P_{act}$. For adaptive repair, it is replaced by the current amount of active repair machinery $\tilde{\beta} P_{act} = P_{r,a}$, which is produced at a certain rate, which is a fraction $\hat{\beta}$ of growth, calculated for each individual at every time step depending on its current fraction of damage (age $Z$) from the following equation:

$$\hat{\beta} = \left(\frac{Z}{1-Z}\right) \max\left(\pm \sqrt{\frac{Y_r}{\mu(S)} \frac{1}{1-Z}} - 1\right) \tag{3}$$

where $\hat{\beta}$ is the value of $\beta$ that maximizes the rate of active protein production and is derived from $\frac{d(dP_{act}/dt)}{d\beta} = 0$.

For $\mu(S)$ in equation 3, we do not take the gross specific growth rate ($\mu_G$) according to equation 1 but the net specific growth rate ($\mu_N$) of each individual cell calculated from its change of total mass from

one iteration to the next, which, due to age and inefficient repair, could be less than the gross specific growth rate.

This gives the following differential equations for the four components of each individual cell for the case of toxic damage that is being repaired:

$$\frac{d}{dt}P_{g,a} = (1 - \tilde{\beta})\,\mu(S)\,P_{g,a}(1 - Z) - A\,P_{g,a} + Y_r P_{r,a}\frac{P_{g,d}}{P_{r,a} + P_{r,d}} \tag{4a}$$

$$\frac{d}{dt}P_{r,a} = \tilde{\beta}\,\mu(S)\,P_{g,a}(1 - Z) - A\,P_{r,a} + Y_r P_{r,a}\frac{P_{r,d}}{P_{r,a} + P_{r,d}} \tag{4b}$$

$$\frac{d}{dt}P_{g,d} = A\,P_{g,a} - P_{r,a}\frac{P_{g,d}}{P_{r,a} + P_{r,d}} \tag{4c}$$

$$\frac{d}{dt}P_{r,d} = A\,P_{r,a} - P_{r,a}\frac{P_{r,d}}{P_{r,a} + P_{r,d}} \tag{4d}$$

where $A$ is a placeholder for the use of the constant, $a$, or gross specific growth rate-proportional damage accumulation rate, $a[\mu(S)]$, calculated as follows:

$$a(\mu) = a'\,\mu(S)$$

**(iii) Cell division.** Upon cell division, the postdivision protein masses of the old pole cell are as follows:

$$P_{a,g} = (1 - \theta)P'_{a,g} - \alpha\theta P'_{d,g} \tag{5a}$$

$$P_{a,r} = (1 - \theta)P'_{a,r} - \alpha\theta P'_{d,r} \tag{5b}$$

$$P_{d,g} = (1 - \theta)P'_{d,g} + \alpha\theta P'_{d,g} \tag{5c}$$

$$P_{d,r} = (1 - \theta)P'_{d,r} + \alpha\theta P'_{d,r} \tag{5d}$$

and those of the new pole cell are as follows:

$$P_{a,g} = \theta P'_{a,g} + \alpha\theta P'_{d,g} \tag{6a}$$

$$P_{a,r} = \theta P'_{a,r} + \alpha\theta P'_{d,r} \tag{6b}$$

$$P_{d,g} = \theta P'_{d,g} - \alpha\theta P'_{d,g} \tag{6c}$$

$$P_{d,r} = \theta P'_{d,r} - \alpha\theta P'_{d,r} \tag{6d}$$

where the prime indicates the protein amounts in the predivision cell, $\theta$ the proportion of protein inherited by the new pole cell, and $\alpha$ the asymmetry of cell division ($\alpha = 1$ fully asymmetric division; $\alpha = 0$ fully symmetric divisions). However, if there is more damage than the old pole daughter cell can take, e.g., when $(1 - \theta)\,P'_{a,g} < \alpha\theta P'_{d,g}$ or $(1 - \theta)\,P'_{a,r} < \alpha\theta P'_{d,r}$, such that the old pole cell would inherit a negative quantity of active protein following equation 5, the old pole cell is assumed to be filled with damaged protein:

$$P_{a,g} = 0 \tag{7a}$$

$$P_{a,r} = 0 \tag{7b}$$

$$P_{d,g} = (1 - \theta)(P'_{a,g} + P'_{d,g}) \tag{7c}$$

$$P_{d,r} = (1 - \theta)(P'_{a,r} + P'_{d,r}) \tag{7d}$$

and the new pole cell inherits all of the active protein, plus the remainder of the damaged protein:

$$P_{a,g} = P'_{a,g} \tag{8a}$$

$$P_{a,r} = P'_{a,r} \tag{8b}$$

$$P_{d,g} = \theta P'_{d,g} - (1 - \theta)P'_{a,g} \tag{8c}$$

$$P_{d,r} = \theta P'_{d,r} - (1 - \theta)P'_{a,r} \tag{8d}$$

**(iv) Damage accumulation (aging).** The growth-independent damage accumulation rate of $0.1\ h^{-1}$, used as described previously by Clegg et al. (45), was suitable for constant and chemostat (spatially uniform, steady-state) environments but was not suitable for biofilm (spatially structured) environments. Most cells within biofilms are not actively growing. Only those cells present in the top or active layer of biofilms have access to substrate and are actively growing. Damage is primarily a by-product of metabolism such as protein folding and respiration; it therefore makes sense to assume that nongrowing cells that do not produce proteins or respire do not accumulate damage (19, 75, 76). Thus, the damage accumulation rate was assumed to be proportional to the cellular specific growth rate.

In order to compare strategies across environments, we need to apply the same damage accumulation rate in all three environments. Hence, the new damage accumulation rate, $a'$, which is proportional to the gross specific growth rate, must be determined to match the fixed damage accumulation rate in the spatially uniform environments ($a = 0.1\ h^{-1}$) where the specific growth rate is constant or predictable for a steady state. In the steady state of the chemostat, the net specific growth rate, $\mu_N$, is equal to the dilution rate, $D$, which was set to $0.3\ h^{-1}$. In iDynoMiCS, the gross specific growth rate, $\mu_G$, is calculated with the Monod equation and depends on substrate concentration, $S$. However, how much higher the gross specific growth rate has to be than the net specific growth rate depends on the age of

the cell, its rate of repair, and its current level of investment into repair. It is therefore difficult to work out analytically; thus, we had to run a number of simulations with different ratios of damage accumulation rates to specific growth rates to find that a value of 0.22 h$^{-1}$/$\mu_G$ would match the previously used constant damage accumulation rate for chemostats (Fig. S5).

**(v) Diagram of processes.** A diagram containing a brief overview of all cellular processes is presented in Fig. 1.

**iDynoMiCS simulations. (i) Cell strategies.** Six combinations of damage segregation and repair strategies were used in this study, but only the fittest three of these were used in biofilm simulations: asymmetric division without repair (DS), symmetric division with adaptive repair (AR), and symmetric division with fixed repair (FR). They are described in the schematic depicted in Fig. 1. Damage is considered to be toxic in all simulations as toxic damage leads to greater differences between strategies (45).

**(ii) Comparison with previous fixed-repair strategy.** In order to compare the new, adaptive repair strategy developed here with the previous fixed-repair strategy (as described in reference 45), single strategy and competition simulations were run in iDynoMiCS. These simulations were initiated with 1,000 cells for single strategies or 500 cells for each strategy for competitions. The single-strategy simulations were run for 3 days in the constant environment, where the "old pole" cell was artificially kept in the simulation, rather than allowing random removal, to examine the consequences of adaptive repair for an individual cell (Fig. 2). Alternatively, they were run for 500 days, to examine the consequences of adaptive repair for populations of cells. Competitions between cells corresponding to different strategies were run in both the constant and chemostat environments for a maximum of 500 days, or until only one strategy remained in the simulation domain (n = 10 for each).

**(iii) Biofilm environment simulations.** We initially carried out biofilm simulations with a single strategy without damage accumulation or repair to determine the parameters that would give rise to typical biofilm structures. For these, cells were initially placed equidistantly on the substratum surface. For simulations with damage accumulation and repair, cells were initially placed randomly on the substratum surface.

In order to examine the effect of several key parameters on the fitness ranking of strategies for cells growing in biofilms, we carried out additional simulations, in triplicate, with two additional values, i.e., one value higher and one value lower than the original parameter value used (see Table S1B in the supplemental material). The parameters varied were as follows: (i) maximum specific growth rate, $\mu_{max}$; (ii) biomass growth yield, $Y_\mu$; (iii) yield of repair, $Y_r$; (iv) division threshold, $P_{div}$; (v) substrate concentration in the bulk liquid, $S_{bulk}$; (vi) substrate affinity, $K_S$; and (vii) proportional aging rate, $a'$. The values used for these are based on the highest and lowest values available in the literature, mostly for *E. coli*; some values are outside what is typical but represent physically possible results (Table S1B). Simulations were carried out at only the highest initial cell density of 32 cells (16 cells for each strategy), and cells were initially placed at random on the substratum surface. Simulations were run for a maximum of 500 h but, in the case of the division threshold, $P_{div}$, were stopped sooner due to computational limitations with huge numbers of small cells. In order to further understand the dynamics that led to damage segregation being a fitter strategy than adaptive repair in specific cases, further simulations were run using only the higher and lower values for $S_{bulk}$, $K_S$, and $a'$; cells were placed in two side-by-side blocks with the cells evenly spaced on the substratum surface. These were run for a maximum of 100 h. The same side-by-side setup where competing strategies are spatially segregated was again used for simulations with higher $S_{bulk}$ values combined with either styrofoam production (where, instead of volume being lost due to the inefficiency of repair, the volume is converted to an inert material type dubbed "styrofoam") or 100% effective repair ($Y_r = 1$).

**Software and hardware used.** iDynoMiCS 1.5 is free open source software written in Java (45, 64). Analysis scripts were written in Python 3.7 (Python Software Foundation, 2010). All source and analysis code can be found at https://github.com/R-Wright-1/iDynoMiCS_1.5 along with example protocol files and instructions on how to repeat the analyses performed here.

**Determining the prevalence and abundance of genes encoding repair functions in microbial genomes.** To investigate the ubiquity of repair functions in prokaryotes, the 10 prokaryotes with the largest number of assemblies available in the NCBI database were determined (Table S1D). These are mostly well-studied model organisms. The annotations of representative genomes of each of these "model" species were obtained from the IMG genome database (72). We also obtained a list of the KEGG orthologs (102) in a database of 20,000 bacterial genomes from the IMG database (72). We then determined the prevalence and abundance of KEGG orthologs known to be associated with repair (Table S1C) in all of these genomes. Where these genomes did not contain any of the genes associated with repair, we obtained the genomes of these organisms from the IMG genome database (72) and used Prokka (103) and CheckM (104) to annotate and determine the quality of these genomes, respectively.

## SUPPLEMENTAL MATERIAL

Supplemental material is available online only.

**TEXT S1**, DOCX file, 0.02 MB.

**FIG S1**, TIF file, 1.5 MB.

**FIG S2**, TIF file, 0.4 MB.

**FIG S3**, TIF file, 1 MB.

**FIG S4**, TIF file, 1.3 MB.

**FIG S5**, TIF file, 0.7 MB.
**FIG S6**, TIF file, 2 MB.
**FIG S7**, TIF file, 1.9 MB.
**FIG S8**, TIF file, 1.6 MB.
**TABLE S1**, DOCX file, 0.03 MB.

## ACKNOWLEDGMENTS

R.J.W. and T.L.R.C. were supported by the Biotechnology and Biological Sciences Research Council (BBSRC), United Kingdom; R.J.W. via a Midlands Integrative Biosciences Training Partnership Ph.D. scholarship; and T.L.R.C. via a University of Warwick Systems Biology Doctoral Training Centre Ph.D. scholarship. R.J.C. and J.-U.K. are grateful to the United Kingdom National Centre for the Replacement, Refinement & Reduction of Animals in Research (NC3Rs) for funding their development of individual-based models (IBMs) for the gut environment (eGUT grant NC/K000683/1). The funders had no role in study design, data collection and interpretation, or the decision to submit the work for publication.

J.-U.K. and R.J.C. initially designed the study with later input from R.J.W. and T.L.R.C. All simulations and analyses whose results are shown were carried out by R.J.W. with guidance from R.J.C. and J.-U.K. T.L.R.C. performed preliminary simulation experiments to choose many of the simulation parameters. R.J.W. and J.-U.K. wrote the first draft of the manuscript, and all of us contributed to revisions.

We declare that we have no conflicts of interest.

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
