## [Reviewer comments · mSystems]

Damage repair versus aging in an individual-based model of biofilms

Robyn Wright, Robert Clegg, Timothy Coker, and Jan-Ulrich Kreft

Corresponding Author(s): Jan-Ulrich Kreft, University of Birmingham

Review Timeline:

Submission Date:	January 7, 2020
Editorial Decision:	February 25, 2020
Revision Received:	April 29, 2020
Editorial Decision:	May 27, 2020
Revision Received:	August 27, 2020
Accepted:	September 16, 2020

Editor: Vanni Bucci

Reviewer(s): Disclosure of reviewer identity is with reference to reviewer comments included in decision letter(s). The following individuals involved in review of your submission have agreed to reveal their identity: Christian Diener (Reviewer #1)

Transaction Report:

DOI: <https://doi.org/10.1128/mSystems.00018-20>

February 25, 2020

Dr. Robyn J Wright
Dalhousie University
School for Resources and Environmental Studies
Halifax, Nova Scotia
Canada

Re: mSystems00018-20 (Damage repair versus aging in biofilms)

Dear Dr. Robyn J Wright:

Thank you for submitting the manuscript to mSystems. Both myself and the reviewer find your paper interesting. Before acceptance of the manuscript I recommend that you address all the concerns raised by the reviewer.

Below you will find the comments of the reviewers.

To submit your modified manuscript, log onto the eJP submission site at <https://msystems.msubmit.net/cgi-bin/main.plex>. If you cannot remember your password, click the "Can't remember your password?" link and follow the instructions on the screen. Go to Author Tasks and click the appropriate manuscript title to begin the resubmission process. The information that you entered when you first submitted the paper will be displayed. Please update the information as necessary. Provide (1) point-by-point responses to the issues raised by the reviewers as file type "Response to Reviewers," not in your cover letter, and (2) a PDF file that indicates the changes from the original submission (by highlighting or underlining the changes) as file type "Marked Up Manuscript - For Review Only."

Please return the manuscript within 60 days; if you cannot complete the modification within this time period, please contact me. If you do not wish to modify the manuscript and prefer to submit it to another journal, please notify me of your decision immediately so that the manuscript may be formally withdrawn from consideration by mSystems.

To avoid unnecessary delay in publication should your modified manuscript be accepted, it is important that all elements you upload meet the technical requirements for production. I strongly recommend that you check your digital images using the Rapid Inspector tool at <http://rapidinspector.cadmus.com/RapidInspector/zmw/>.

Sincerely,

Vanni Bucci

Editor, mSystems

Journals Department
Reviewer comments:

Reviewer #1 (Comments for the Author):

The authors present results from a mathematical model to study the impact of different cellular damage repair strategies on growth rates in spatially homogeneous ("chemostat") and heterogeneous environments ("biofilm"). Fitness benefits between damage mitigation strategies were quantified with competition setups between pairs of simulated cellular populations. The authors conclude that cellular resource investment into damage repair always infers a fitness benefit over damage segregation during cell division.

Evaluation

I think the presented ideas are interesting and the strategy to simulate competition experiments between different damage repair strategies to infer differential fitness seems appealing to me. I did find the article a bit difficult to read. In particular, the article does not stress enough that it only presents simulation results, gives only very little motivation and explanation for the used modeling strategy and does not attempt to validate any of the presented simulations. It was also unclear to me how aging was included in the mathematical model.

Suggested major changes

1. I think the article would benefit greatly from a better introduction and motivation for the used modeling strategy. Even the abstract contains about 3/4 of background on damage repair and only 1/4 of description of what was done in the paper. I would recommend giving more space to the mathematical modeling in the title, abstract and manuscript. For instance, it seems like the authors use an agent based model with individual-based ODEs in each agent but that is not mentioned anywhere in the main text and not even clearly stated in the methods section. I think it would also help to have a general description of the modeling and simulation strategy in Fig. 1 (maybe simplifying the existing Fig. 1 as well). It was also unclear to me which biological system the authors are trying to simulate (and which assumptions they make to that end). Mechanisms of damage repair vary between and across pro- and eukaryotes so I would expect the model to describe some organisms better than others.

2. The manuscript only presents simulation results from a complex model with many free parameters and shows results under a single set of chosen parameters. I feel that this lacks some kind of validation against real data in order to evaluate how well the model would recapitulate a bacterial biofilm. At the least I would expect some robustness analysis of the parameters to see if the results remain the same under variation of the chosen parameters.

3. It is unclear how aging was treated in the simulations. From Fig. 1 it seems like it was integrated as a general accumulation rate of damaged protein with age being defined as the amount of accumulated damage (lines 207-211). The concept of aging is not well defined for prokaryotes and is often quantified by replicative or chronological lifespan which differs from aging in complex eukaryotes where damage is related to some but not all hallmarks of aging (<https://dx.doi.org/10.1016/j.cell.2013.05.039>). It would be helpful if the manuscript provided a clear definition of what the authors consider aging in their mathematical modeling and how that fits into the existing literature. It is also unclear to me what the particular impact of bacterial aging in biofilms would be. Have there been experimental observations that biofilms have altered aging characteristics of that aging is different in bacteria that are prone to forming biofilms?

Suggested minor changes

1. The introduction should give a better definition of which types of damage are considered here, for instance does the model treat DNA damage and associated mutations which are always inherited to the offspring?

2. Looking at references 1-3 it is unclear to me how they support the statements made in lines 75-80.

3. "Ironically" should be substituted with "Strikingly" or "Surprisingly".

4. The phrase in lines 138-139 ("For humans..") needs references.

5. Some of the terms in the paper should be defined in the text or in a glossary. For instance "unicells" (prokaryotes?) and "cellular specific growth rate" (single cell/individual growth rate?).

6. The phrase in lines 222-223 needs a reference.

7. Fig 1. Would be clearer if there were arrows between the 2 division strategies and the 3 repair strategies labeled by the roman number of the specific combination.

8. Fig. 3: indicate the numerator and denominator of the "log biomass ratio" on the axis and caption. Describe what the colors mean.

9. The study would be much more reproducible if the authors provided some description to the files linked in line 803. The Github has no "README" file and gives no indication what would need to be done to reproduce the simulations in the manuscript.

10. I would suggest splitting some of the longer phrases into shorter ones to improve legibility.

Reviewer comments:

Reviewer #1 (Comments for the Author):

The authors present results from a mathematical model to study the impact of different cellular damage repair strategies on growth rates in spatially homogeneous ("chemostat") and heterogeneous environments ("biofilm"). Fitness benefits between damage mitigation strategies were quantified with competition setups between pairs of simulated cellular populations. The authors conclude that cellular resource investment into damage repair always infers a fitness benefit over damage segregation during cell division.

Thank you for your constructive criticism and the many useful suggestions.

Evaluation

I think the presented ideas are interesting and the strategy to simulate competition experiments between different damage repair strategies to infer differential fitness seems appealing to me. I did find the article a bit difficult to read. In particular, the article does not stress enough that it only presents simulation results, gives only very little motivation and explanation for the used modeling strategy and does not attempt to validate any of the presented simulations. It was also unclear to me how aging was included in the mathematical model.

Thank you for highlighting these issues. They are the unintended consequences of staying within the word limit by avoiding repeating in depth what we explained more thoroughly in our previous publication, Clegg et al. (2014).

We have now highlighted that this study is entirely based on simulations (title, abstract, introduction lines 55, 160-164, results lines 173-175), that the modelling strategy was to capture generic unicellular organisms (avoiding details that are subject to evolution, lines 158-160, 169-173 and 175-177) with alternative 'aging' strategies whose fitnesses are assessed by competition (rather than using arbitrary fitness functions or definitions). Many of these areas, along with definitions of some key terms, are now in an additional paragraph that comes at the beginning of the results section (lines 169-201)

Importantly, because we want to derive generic results, we validate against the entirety of experimental findings on unicellular organisms. These are in a nutshell:

(1) All unicellular (also multicellular) organisms studied have evolved (several) mechanisms of damage repair, validating our prediction that repair is beneficial under all circumstances, now including biofilms.

(2) According to the more rigorous experimental studies based on large numbers of single-cell observations over many generations, damage segregation in unicellular microorganisms does not occur under benign conditions. So aging is not universal, only repair is.

(3) Segregation does occur under stress, where more damage accumulates than can be effectively repaired, so we and other recent studies interpret damage segregation as a stress response, a strategy to deal with damage 'overflow', a plan B for when repair alone does not suffice.

E. coli for example has no active mechanism of damage segregation as you would expect if it were beneficial for fitness, but damage often ends up at the poles due to exclusion of random walking aggregates from the nucleoid region in the cell's centre. As the old pole exists for longer, aggregates are more likely at the older pole. This stochastic 'mechanism' is not very

effective (low degree of asymmetry, not effective for damage that does not form aggregates) and since it is inevitable to happen, it cannot be known whether it evolved due to fitness benefits, or is simply happening because it is difficult to avoid.

(4) There is one exception, the budding yeast, which has active and effective mechanisms of damage segregation and a clearly limited replicative lifespan. Regarding the budding yeast, we argue that it is actually not a unicellular organism as wild strains show hyphal growth, so the domesticated budding yeast's ancestors were multicellular for millions of years. Indeed, a domesticated strain rapidly evolves multicellularity (snowflake yeast). This explains the difference to the truly unicellular fission yeast.

(5) Note that previous mathematical models that predicted damage segregation to be more advantageous than repair, are invalidated by the fact that all studied organisms clearly evolved repair.

What our new study shows is that repair is beneficial also in spatially structured environments. If it were only beneficial in well-mixed environments, it would be hard to understand why repair is universal given how prevalent biofilms are.

See below on how aging was included in the model.

Suggested major changes

1. I think the article would benefit greatly from a better introduction and motivation for the used modeling strategy. Even the abstract contains about 3/4 of background on damage repair and only 1/4 of description of what was done in the paper. I would recommend giving more space to the mathematical modeling in the title, abstract and manuscript. For instance, it seems like the authors use an agent based model with individual-based ODEs in each agent but that is not mentioned anywhere in the main text and not even clearly stated in the methods section. I think it would also help to have a general description of the modeling and simulation strategy in Fig. 1 (maybe simplifying the existing Fig. 1 as well). It was also unclear to me which biological system the authors are trying to simulate (and which assumptions they make to that end). Mechanisms of damage repair vary between and across pro- and eukaryotes so I would expect the model to describe some organisms better than others.

Thank you for making us realize that we have not made some crucial points clear enough. Prompted by word limits, we thought we should not flesh out what we already made clear in the previous study. We have now made numerous additions to clarify the points raised without expecting the reader to read the previous study first (incorporated into the new paragraph at the beginning of the results section, lines 169-201:

Motivation of modelling strategy.

Characterization of the model (individual agents have ODEs inside).

That we aim to simulate a generic unicellular organism as details of repair and damage segregation are evolvable traits and differ between unicells, as the reviewer has pointed out. We want to avoid making assumptions on details that could differ between unicells – it is not a model of *E. coli* or any other organism. In some places we use rates that were determined for *E. coli* simply because this is in many instances one of the only organisms that these have been determined for.

The only suggestion we find hard to follow is to include more results in the abstract. We were unhappy to have little concrete results in the abstract ourselves and tried to include them, but due to the complexities of the results we found it impossible to give a more detailed account within the word limit and a compromise of including some details but not others

would have been misleading. We considered cutting the introductory part in the abstract, but felt that we needed to introduce the problem very clearly for the rest to make any sense.

2. The manuscript only presents simulation results from a complex model with many free parameters and shows results under a single set of chosen parameters. I feel that this lacks some kind of validation against real data in order to evaluate how well the model would recapitulate a bacterial biofilm. At the least I would expect some robustness analysis of the parameters to see if the results remain the same under variation of the chosen parameters.

Regarding validation against real data: There are no experimental studies of aging in biofilms which precludes direct validation against experimental data. This is one key reason why our strategy has been to validate our model predictions qualitatively against the entirety of evidence available. If biofilms would select against repair, and most unicells live in biofilms most of the time, there should be unicells that do not repair. As these have not been found, our predictions for biofilms are consistent with evidence and likely to be true.

Another key reason for 'abstract validation' was our aim to make generic predictions for all unicells, so we wanted to avoid making a model of a specific organism and a specific biofilm. Specifics will have changed in the course of evolution of unicells over billions of years.

Regarding robustness analysis, we have done a more thorough job in our first study of constant and chemostat environments, where it was feasible to run thousands of replicates for cases where fitness differences were small. Even there, we had to limit the analysis to aging relevant parameters rather than general growth kinetic parameters such as substrate affinity (not relevant in constant environments). Given the larger variation in results from biofilm simulations, we restricted our analysis to toxic damage in the current study (if damage is not toxic, fitness differences are very small but the relative ranking of strategies was the same in all cases in Clegg et al. 2014, apart from tiny damage accumulation rates). Moreover, we did not repeat other items of the robustness analysis in the previous study because we had found that, for example, intermediate asymmetry lead to intermediate outcomes, or increased (or decreased) repair efficiency made repair more (or less) beneficial. That is, changes in parameters did not affect qualitative outcomes – the ranking of strategies. And that is what our study really is about, fitness ranking of strategies. We did, however, examine some key parameters: initial density of cells in biofilms, well-mixed environments vs spatially structured environments and parameters relating to substrate diffusion in biofilms (simulations without damage accumulation or repair).

We have now made these points clearer in the manuscript.

3. It is unclear how aging was treated in the simulations. From Fig. 1 it seems like it was integrated as a general accumulation rate of damaged protein with age being defined as the amount of accumulated damage (lines 207-211). The concept of aging is not well defined for prokaryotes and is often quantified by replicative or chronological lifespan which differs from aging in complex eukaryotes where damage is related to some but not all hallmarks of aging (<https://dx.doi.org/10.1016/j.cell.2013.05.039>). It would be helpful if the manuscript provided a clear definition of what the authors consider aging in their mathematical modeling and how that fits into the existing literature. It is also unclear to me what the particular impact of bacterial aging in biofilms would be. Have there been experimental observations that biofilms have altered aging characteristics of that aging is different in bacteria that are prone to forming biofilms?

Yes, aging is implemented as rate of accumulation of damage in the model, where 'protein' is meant to represent biomass in general (protein is just a name for the whole of the biomass, again something we failed to make clear enough as it was made clear in Clegg et al. 2014, lines 184-185). Thus, age is defined not as chronological age or number of divisions but as the fraction of damage. The effect of aging or age, declining cellular function (growth or survival) is better called senescence and we have now gone through the manuscript to ensure we use the more specific term senescence where we talk about deterioration with age, and damage accumulation when we talk about the aging implemented in our model. Much of the literature uses 'aging' as a fairly vague umbrella term that can have various meanings and then the more specific terms such as chronological age, replicative lifespan, mortality, fecundity and senescence, with various definitions. We have checked that we define all terms that we use clearly (lines 77-78) and now point out that there are other definitions in the literature (lines 180-185). We intend to cover this more thoroughly in a review that is in preparation. Regarding biofilms, there are no experimental observations. We expect the impact of aging to be twofold, one is changes in growth parameters (lower specific growth rate of older cells or cells that invest more into repair and less into growth machinery, reduced yield due to repair requiring resources). The other impact is that asymmetric division leads to a mixture of young and old cells in the same position within the biofilm (until lack of growth precludes further divisions and this demixing). We explain this now. Thank you for helping us improve.

Suggested minor changes

1. The introduction should give a better definition of which types of damage are considered here, for instance does the model treat DNA damage and associated mutations which are always inherited to the offspring?

Thank you for raising this. We do not consider mutations, as these are very different, being heritable and subject to natural selection, whereas natural selection does not act on the accumulation of misfolded proteins or passive, inevitable damage segregation. Selection can act on active mechanisms to prevent, repair or segregate damage. We made this clear now (lines 185-188).

2. Looking at references 1-3 it is unclear to me how they support the statements made in lines 75-80.

Thank you for pointing this out. We have now ensured that the other references are used at the correct point in the text, rather than all at the end of the sentence (now lines 77-84).

3. "Ironically" should be substituted with "Strikingly" or "Surprisingly".

We have now replaced "Ironically" with "Surprisingly".

4. The phrase in lines 138-139 ("For humans..") needs references.

We thank the reviewer for noticing this and have now added references to this line (now lines 143-144).

5. Some of the terms in the paper should be defined in the text or in a glossary. For instance "unicells" (prokaryotes?) and "cellular specific growth rate" (single cell/individual growth rate?).

We have now defined unicells as unicellular microorganisms (lines 86-87, 169-170) and pointed out that this includes prokaryotic and eukaryotic cells as the presence or absence of a nucleus is not relevant for aging. Whether there is a clear germline soma distinction as in multicellular organisms or not matters crucially.

6. The phrase in lines 222-223 needs a reference.

We thank the reviewer for noticing this and have now added a reference to this line (now lines 260-263).

7. Fig 1. Would be clearer if there were arrows between the 2 division strategies and the 3 repair strategies labeled by the roman number of the specific combination.

We have now modified and simplified Figure 1 and believe that this is now much easier to understand. We prefer not to add the suggested roman numerals as we believe that this would be a misrepresentation as each strategy that we investigate is a combination of the repair and division strategies, a point that we now make much clearer in the legend.

8. Fig. 3: indicate the numerator and denominator of the "log biomass ratio" on the axis and caption. Describe what the colors mean.

We have now added description to the figure legend of what the colors mean as well as which log biomass ratios are referred to in each panel of Figure 3. This also made us realise that we had also not made this clear in Figure 6, so we have now also added this to the legend of Figure 6.

9. The study would be much more reproducible if the authors provided some description to the files linked in line 803. The Github has no "README" file and gives no indication what would need to be done to reproduce the simulations in the manuscript.

We thank the reviewer for pointing this out and we have now added a README file to the Github repository that details how the analyses in this manuscript would be reproduced as well as which example simulations relate to which sections of the manuscript. We have also ensured that all example simulations are thoroughly commented to show the parameters that we vary in this manuscript. We also added Figure S1 to summarise the aims of our study, the simulations that we have carried out here, the parameters that are varied in each section, which figures these lead to and the key results of each manuscript section.

10. I would suggest splitting some of the longer phrases into shorter ones to improve legibility.

Thank you, we have gone through the manuscript checking for long phrases and split them up where possible.

May 27, 2020

Dr. Jan-Ulrich Kreft
University of Birmingham
Centre for Systems Biology
School of Biosciences
Birmingham
United Kingdom

Re: mSystems00018-20R1 (Damage repair versus aging in an individual-based model of biofilms)

Dear Dr. Jan-Ulrich Kreft:

The reviewer has concerns with experimental validation. Being a modeler myself, I understand it may not always be possible to reproduce or produce exact experimentally-measured quantities. I find value in qualitative model-produced results. I would therefore ask to revisit the text and be more specific about what experimental observations the model is capturing.

Below you will find the comments of the reviewers.

To submit your modified manuscript, log onto the eJP submission site at <https://msystems.msubmit.net/cgi-bin/main.plex>. If you cannot remember your password, click the "Can't remember your password?" link and follow the instructions on the screen. Go to Author Tasks and click the appropriate manuscript title to begin the resubmission process. The information that you entered when you first submitted the paper will be displayed. Please update the information as necessary. Provide (1) point-by-point responses to the issues raised by the reviewers as file type "Response to Reviewers," not in your cover letter, and (2) a PDF file that indicates the changes from the original submission (by highlighting or underlining the changes) as file type "Marked Up Manuscript - For Review Only."

Due to the SARS-CoV-2 pandemic, our typical 60 day deadline for revisions will not be applied. I hope that you will be able to submit a revised manuscript soon, but want to reassure you that the journal will be flexible in terms of timing, particularly if experimental revisions are needed. When you are ready to resubmit, please know that our staff and Editors are working remotely and handling submissions without delay. If you do not wish to modify the manuscript and prefer to submit it to another journal, please notify me of your decision immediately so that the manuscript may be formally withdrawn from consideration by mSystems.

To avoid unnecessary delay in publication should your modified manuscript be accepted, it is important that all elements you upload meet the technical requirements for production. I strongly recommend that you check your digital images using the Rapid Inspector tool at <http://rapidinspector.cadmus.com/RapidInspector/zmw/>.

Sincerely,

Vanni Bucci

Editor, mSystems

Journals Department
Reviewer comments:

Reviewer #1 (Comments for the Author):

Although the authors now have improved the introduction and abstract I still feel that some more rigorous validation would be necessary.

In particular the authors state that "... because we want to derive generic results, we validate against the entirety of experimental findings on unicellular organisms..." and continue to list 5 observations. I think this is problematic. The model will obviously not reproduce "the entirety of experimental findings on unicellular organisms", even less though with a single set of chosen parameters. I do understand that the authors aim only to generate qualitative agreement but even this would require some more rigorous validation I feel. Simply stating that the model reproduces some very general observation (that [adaptive] damage repair is beneficial) does not necessarily mean the model or any more detailed predictions derived from it are correct.

I agree with the authors that it may be hard to reproduce the entire model setup in an experiment, but would rather suggest to validate the part of the model where data is available. Even though there may be no data on organisms without damage repair, there is plenty of data on growth dynamics in biofilms, for instance recent ones such as <https://doi.org/10.1038/s41567-018-0356-9> or <https://doi.org/10.1038/s41467-020-15165-4>. If the model could reproduce the growth dynamics presented here this would help. I also looked through the original publication introducing the modeling framework (iDynoMiCS, reference 64) but could find no quantitative validation with experimental data there either. So I think it may be necessary here. Showing that the model reproduces the experimental growth dynamics and is impervious to variation of model parameters would make the claims made in the manuscript much stronger.

The authors state that "We did, however, examine some key parameters: initial density of cells in

biofilms, well-mixed environments vs spatially structured environments and parameters relating to substrate diffusion in biofilms (simulations without damage accumulation or repair).

We have now made these points clearer in the manuscript."

This statement did not include any line numbers and I could not find any changes made in the text to that effect. In particular, I am still missing a broader sampling of the key parameters listed in Table 1 such as growth rate, damage accumulation rates, or biomass required for cell division. All of those quantities vary by several orders of magnitude across different microbes and just substituting them by representative values is unlikely to yield as generalizable results as the authors claim.

Minor changes:

I would recommend including the formulation "mathematical model" or "computational model" in the abstract.

Lines 143-145. Putting a number on this prevalence might make the argumentation stronger. <https://doi.org/10.1038/s41579-019-0158-9> provides a good discussion of that.

Reviewer comments:

Reviewer #1 (Comments for the Author):

We again thank the reviewer for their time in reviewing our manuscript and their helpful and constructive comments. In particular, thank you for making us do the sensitivity analysis including growth parameters. We did not think it would make any difference but we were wrong and it did. We found two situations where DS wins, and took some time to figure out why. It turned out to be a combination of two mechanisms with one environmental condition that are required for DS to win. There are two other effects that are present but that do not explain why DS wins when it does. This has led to more substantial additions and changes in many places and will make the conclusions more reliable and robust. Thank you again.

Regarding evidence for our model prediction that repair is advantageous, which is in contrast with all other models published before our first paper, we thought it was an established fact that all (known) organisms have repair mechanisms for damaged cell material so we thought there was no need to go into detail. We have now searched 20,000 genomes and found a handful that did not have any annotated repair mechanism, but there are explanations for these exceptions.

Although the authors now have improved the introduction and abstract I still feel that some more rigorous validation would be necessary.

In particular the authors state that "... because we want to derive generic results, we validate against the entirety of experimental findings on unicellular organisms..." and continue to list 5 observations. I think this is problematic. The model will obviously not reproduce "the entirety of experimental findings on unicellular organisms", even less though with a single set of chosen parameters. I do understand that the authors aim only to generate qualitative agreement but even this would require some more rigorous validation I feel. Simply stating that the model reproduces some very general observation (that [adaptive] damage repair is beneficial) does not necessarily mean the model or any more detailed predictions derived from it are correct.

Our wording 'validate against the entirety of experimental findings' was poor. What we were trying to explain is that our generic model makes qualitative and general predictions that should be validated against all relevant empirical knowledge from all unicells. Basically, our model shows that repair of damage is advantageous for the fitness of unicells under a wide range of conditions leading to the expectation that repair should have evolved in unicells and should be active in all unicells. Other models found the opposite. Thus our model would be supported (maybe a better word than validated) by experimental evidence of repair mechanisms occurring in unicells and any unicells without repair mechanisms would support the other models. We thought it was a well established fact that unicells have mechanisms of damage repair (in some model organisms these have been studied extensively) so we did not provide details of this evidence. Given the confusion this has generated, we have now decided that it would help to clarify what it means to have empirical support for the prediction of our generic model by providing a detailed analysis of the occurrence of repair mechanisms in unicells.

So we have now carried out searches of the genomes of 20,000 microbial isolates as well as the genomes of the ten 'model' prokaryotes with the largest number of assemblies on the NCBI database. We have downloaded the annotations of these genomes from the JGI Integrated Microbial Genome database and have used these to determine the prevalence and abundance of KEGG orthologs related to repair (excluding DNA repair that we do not consider in our model).

Almost all genomes contain repair genes and those that don't are incomplete. Full details of these methods and results of these searches can be found in lines 486-501 (results), 1002-1013 (methods) and Tables S1C-S1E, Figure S8 and in the new supplementary figshare file at <https://doi.org/10.6084/m9.figshare.12515526>.

I agree with the authors that it may be hard to reproduce the entire model setup in an experiment, but would rather suggest to validate the part of the model where data is available. Even though there may be no data on organisms without damage repair, there is plenty of data on growth dynamics in biofilms, for instance recent ones such as <https://doi.org/10.1038/s41567-018-0356-9> or <https://doi.org/10.1038/s41467-020-15165-4>. If the model could reproduce the growth dynamics presented here this would help. I also looked through the original publication introducing the modeling framework (iDynoMiCS, reference 64) but could find no quantitative validation with experimental data there either. So I think it may be necessary here. Showing that the model reproduces the experimental growth dynamics and is impervious to variation of model parameters would make the claims made in the manuscript much stronger.

The authors state that "We did, however, examine some key parameters: initial density of cells in biofilms, well-mixed environments vs spatially structured environments and parameters relating to substrate diffusion in biofilms (simulations without damage accumulation or repair).

We have now made these points clearer in the manuscript."

This statement did not include any line numbers and I could not find any changes made in the text to that effect. In particular, I am still missing a broader sampling of the key parameters listed in Table 1 such as growth rate, damage accumulation rates, or biomass required for cell division. All of those quantities vary by several orders of magnitude across different microbes and just substituting them by representative values is unlikely to yield as generalizable results as the authors claim.

It is true that there are experimental studies of growth dynamics of biofilms, but validating our model against these studies would not help decide whether our model prediction that repair is advantageous is more correct than the predictions of alternative models on aging. What would be needed is studies with repair defective mutants for example. The papers mentioned are not providing information on the aging question and simply redoing these studies is not our aim. We can't do better than the work of the Drescher lab, Hartmann *et al.* (2019) (1), but their question was a different one (to what extent can mechanics rather than substrate concentration gradients explain biofilm structure formation), repair or damage rate were not varied, and the parameterization is specific to *Vibrio*. Their model captures the biofilm development accurately for the early phase of biofilm growth which is

only driven by mechanical interactions, not substrate diffusion gradients, which forms the basis of our biofilm model. It is the dependence of growth on substrate concentration, and the dependence of damage accumulation on growth rate, that are our concern.

The second paper mentioned by the reviewer, Paula *et al.* (2020) (2) studies biofilm formation in a flow cell similar to our model, but during the time of observation, they suggest that the biofilm growths follows a power law, which is an observation without clear physical interpretation and may come from lumping various growth behaviours in individual microcolonies. Note the range for the exponent of the power law is about 1 to 5. It is hard to see a physical reason for these exponents. In the type of biofilm model we use, based on diffusion reaction equation, you first have exponential growth during the early phase of biofilm growth, before substrate limitation sets in when growth of the biofilm becomes linear. As our interest is in comparing systems with substrate concentration gradients versus chemostats without gradient, this study is not directly useful for our purposes. Our model and others do not give such power law behaviour and this is probably related to the fact that Paula *et al.* (2020) work on a larger scale of many microcolonies.

As mentioned above, we have now run some additional simulations where we vary the values used for key growth parameters including the ones mentioned by the reviewer: 1) maximum specific growth rate, μ_{max} ; 2) biomass growth yield, Y_{μ} ; 3) yield of repair, Y_r ; 4) division threshold (biomass required for cell division), P_{div} ; 5) substrate concentration in the bulk liquid, S_{bulk} ; 6) substrate affinity, K_S ; and 7) proportional aging rate (damage accumulation rate), a' . For each parameter we have performed simulations with the original value, a lower value and a higher value – with each of these being based on experimentally measured values, where possible. Due to the large number of total combinations of these parameters we have only varied each parameter in isolation. Most of the results were as expected, but we found one set of environmental conditions where repair is not advantageous. Full details of the additional simulations performed as well as these results can be found in lines 399-484 (results), 981-1000 (methods), Figures 8, 9, S6 and S7 and in the new supplementary figshare file at <https://doi.org/10.6084/m9.figshare.12515526>.

Minor changes:

I would recommend including the formulation "mathematical model" or "computational model" in the abstract.

We have now added the word "computational" to line 53.

Lines 143-145. Putting a number on this prevalence might make the argumentation stronger. <https://doi.org/10.1038/s41579-019-0158-9> provides a good discussion of that.

We have now added this to lines 144-145.

References

1. Hartmann R, Singh PK, Pearce P, Mok R, Song B, Díaz-Pascual F, Dunkel J, Drescher K. 2019. Emergence of three-dimensional order and structure in growing biofilms. *Nat Phys* 15:251–256.
2. Paula AJ, Hwang G, Koo H. 2020. Dynamics of bacterial population growth in biofilms resemble spatial and structural aspects of urbanization. *Nat Commun* 11:1–14.

September 16, 2020

Dr. Jan-Ulrich Kreft
University of Birmingham
Centre for Systems Biology
School of Biosciences
Birmingham
United Kingdom

Re: mSystems00018-20R2 (Damage repair versus aging in an individual-based model of biofilms)

Dear Dr. Jan-Ulrich Kreft:

Your manuscript has been accepted, and I am forwarding it to the ASM Journals Department for publication. For your reference, ASM Journals' address is given below. Before it can be scheduled for publication, your manuscript will be checked by the mSystems senior production editor, Ellie Ghatineh, to make sure that all elements meet the technical requirements for publication. She will contact you if anything needs to be revised before copyediting and production can begin. Otherwise, you will be notified when your proofs are ready to be viewed.

Sincerely,

Vanni Bucci
Editor, mSystems

Journals Department
Figure S3: Accept

Figure S5: Accept

Supplemental Materials and Methods: Accept

Figure S6: Accept

Figure S2: Accept

Table S1: Accept

Figure S4: Accept

Figure S8: Accept

Figure S7: Accept

Figure S1: Accept